# Impact of 3D Cloud Structures on the Atmospheric Trace Gas Products from UV-VIS Sounders – Part I: Synthetic dataset for validation of trace gas retrieval algorithms

Claudia Emde[1], Huan Yu[3], Arve Kylling[2], Michel van Roozendael[3], Kerstin Stebel[2], Ben Veihelmann[4], and Bernhard Mayer[1]

[1]Ludwig-Maximilians-University (LMU), Meteorological Institute, Munich, Germany
[2]Norwegian Institute for Air Research (NILU), Kjeller, Norway
[3]Belgian Institute for Space Aeronomy (BIRA-IASB), Brussels, Belgium
[4]ESA-ESTEC, Noordwijk, the Netherlands

*Correspondence to:* Claudia Emde (claudia.emde@lmu.de)

**Abstract.** Retrievals of trace gas concentrations from satellite observations are mostly performed for clear regions or regions with low cloud coverage. However, even fully clear pixels can be affected by clouds in the vicinity, either by shadowing or by scattering of radiation from clouds in the clear region. Quantifying the error of retrieved trace gas concentrations due to cloud scattering is a difficult task. One possibility is to generate synthetic data by three-dimensional (3D) radiative transfer simulations using realistic 3D atmospheric input data, including 3D cloud structures. Retrieval algorithms may be applied on the synthetic data and comparison to the known input trace gas concentrations yields the retrieval error due to cloud scattering.

In this paper we present a comprehensive synthetic dataset which has been generated using the Monte Carlo radiative transfer model MYSTIC. The dataset includes simulated spectra in two spectral ranges (400–500 nm and the $O_2A$-band from 755–775 nm). Moreover it includes layer air mass factors (layer-AMF) calculated at 460 nm. All simulations are performed for a fixed background atmosphere for various sun positions, viewing directions and surface albedos.

Two cloud setups are considered: The first includes simple box-clouds with various geometrical and optical thicknesses. This can be used to systematically investigate the sensitivity of the retrieval error on solar zenith angle, surface albedo and cloud parameters. Corresponding 1D simulations are also provided. The second includes realistic three-dimensional clouds from an ICON large eddy simulation (LES) for a region covering Germany and parts of surrounding countries. The scene includes cloud types typical for central Europe such as shallow cumulus, convective cloud cells, cirrus, and stratocumulus. This large dataset can be used to quantify the trace gas concentration retrieval error statistically.

Along with the dataset the impact of horizontal photon transport on reflectance spectra and layer-AMFs is analyzed for the box-cloud scenarios. Moreover, the impact of 3D cloud scattering on the $NO_2$ vertical column density (VCD) retrieval is presented for a specific LES case. We find that the retrieval error is largest in cloud shadow regions, where the $NO_2$ VCD is underestimated by more than 20%.

The dataset is available for the scientific community to assess the behaviour of trace gas retrieval algorithms and cloud correction schemes in cloud conditions with 3D structure.

# 1 Introduction

In order to monitor atmospheric composition, spectra in the UV-Vis spectral range have been observed from space for several decades (Gonzalez Abad et al., 2019) in order to retrieve trace gas concentrations such as ozone ($O_3$) and nitrogen dioxide ($NO_2$). Operational retrieval algorithms employ different methodologies, e.g. the optimal estimation method commonly used to retrieve ozone or water vapor altitude profiles (Rodgers, 2000) or the Differential Optical Absorption Spectroscopy (DOAS) fitting method (e.g., Platt, 2017; Boersma et al., 2018). Many of those algorithms require a cloud-free atmosphere and therefore a cloud mask is often used to filter out all satellite pixels including clouds.

Using radiative transfer models, the influence of various cloud parameters on the retrieval of trace gas columns has been estimated and it was found that cloud fraction, cloud optical thickness and cloud top pressure are the most important quantities (e.g. Boersma et al., 2004; Stammes et al., 2008; Loyola et al., 2018).

Some retrieval algorithms correct for the presence of clouds using radiometric cloud fraction estimates as Stammes et al. (2008), or photon path length correction methods based on $O_2$-$O_2$ (Veefkind et al., 2016) or $O_2$A-band absorption measurements (Loyola et al., 2018; Liu et al., 2021). Three effects have so far been considered in these cloud correction methods: the enhancement of reflectivity compared to clear scenes (albedo effect), the so-called shielding effect (part of the trace gas column is hidden by clouds), and the increase of absorption within the cloud due to enhancement of the photon path-length in the cloud by multiple scattering.

Another effect, which becomes increasingly important with increasing spatial resolution of the instruments, is the impact of three-dimensional (3D) cloud scattering, i.e. cloud shadow effects, enhancement of reflectance by in-scattering of photons from clouds in neighboring pixels, and additional effects by unresolved sub-pixel clouds. For OCO-2 with a spatial footprint of $1.29 \times 2.25\,\text{km}^2$, modelling studies have shown that 3D cloud scattering causes significant biases in $CO_2$ retrievals (Merrelli et al., 2015; Massie et al., 2021). For very high spatial resolution of 2-5 $\text{m}^2$ as achieved with the APEX air-borne spectrometer, 3D effects by cloud scattering and by multiple reflections at buildings have been investigated by Schwaerzel et al. (2020, 2021).

Large deviations are found between the individual cloud retrieval algorithms that are applied for cloud correction (Loyola et al., 2018) in $NO_2$ trace gas retrievals from TROPOMI/S5P observations. In order to evaluate the performance of the retrievals synthetic datasets were used. However, these synthetic datasets were generated using a 1D radiative transfer model, thus 3D cloud scattering effects were not included.

Within the ESA project 3DCATS (Impact of 3D Cloud Structures on the Atmospheric Trace Gas Products from UV-VIS Sounder) we have quantified the $NO_2$ retrieval error due to 3D cloud scattering based on a synthetic data set and on real TROPOMI/S5P observations. We present the results of the project as a series of three publications.

The paper at hand presents the comprehensive synthetic dataset, which has been generated using the 3D radiative transfer model MYSTIC (Mayer, 2009; Emde et al., 2011). The synthetic dataset includes simulated spectra in two spectral ranges (400–500 nm and the $O_2$A-band from 755–775 nm), and layer air mass factors (layer-AMF) calculated at 460 nm. All simulations were performed for a fixed background atmosphere for various sun positions, viewing directions and surface albedos. Two

cloud setups are included, the first is for a simple box-cloud and the second includes realistic clouds from an LES simulation over Europe.

In the second paper by Yu et al. (2021), the sensitivity of the NO$_2$ VCD retrieval error for clear-sky pixels near clouds has been investigated using the synthetic dataset for the box-cloud. Yu et al. (2021) systematically analyze the NO$_2$ VCD retrieval error in terms of the following parameters: solar zenith angle, surface albedo, cloud optical thickness, cloud height, cloud geometrical thickness. This analysis allowed to develop first concepts to correct for 3D cloud scattering in trace gas retrieval algorithms, which were validated using the synthetic dataset including the LES clouds.

In the third paper by Kylling et al. (2021) the NO$_2$ VCD retrieval error due to 3D cloud scattering has been quantified using both synthetic and observational data.

The paper is organized as follows: Section 2 gives a short description of the Monte Carlo radiative transfer model MYSTIC used to generate the sythetic dataset. Section 3 describes the first part of synthetic data including the box-cloud cases, and investigates the impact of 3D cloud scattering on reflectance spectra and layer-AMFs. Section 4 describes the synthetic dataset for realistic LES clouds and demonstrates for a specific case the application of a retrieval algorithm on the synthetic data and the quatification of the NO$_2$ VCD retrieval error. Section 5 includes some concluding remarks.

## 2 Three-dimensional radiative transfer model MYSTIC

The synthetic datasets have been generated using the three-dimensional radiative transfer model MYSTIC (Monte Carlo code for the phYsically correct Tracing of photons in Cloudy atmospheres, Mayer (2009); Emde et al. (2011)). MYSTIC is operated as one of several radiative transfer solvers of the libRadtran software package (www.libRadtran.org, Mayer and Kylling (2005); Emde et al. (2016)). MYSTIC is capable of simulating (polarized) solar and thermal radiances and also irradiances, actinic fluxes, heating rates and box/layer–airmass factors.

MYSTIC has been used extensively to generate synthetic measurements to validate various retrieval algorithms for cloud and aerosol properties (e.g. Kokhanovsky et al., 2010b; Bugliaro et al., 2011; Davis et al., 2013; Stap et al., 2016a, b; Grob et al., 2019). Further application fields are, e.g., photochemistry (Sumińska-Ebersoldt et al., 2012) and remote sensing of exo–planets (Emde et al., 2017). MYSTIC allows the definition of arbitrary complex 3D clouds and aerosols, 2D surface albedo maps and topography. The surface can be treated as Lambertian or by a Bidirectional Distribution Function (BRDF), for both types the spectral dependency may be included. It can be operated in fully spherical geometry (Emde and Mayer, 2007) and is therefore also suitable to simulate limb observations. The polarization state of the radiation can be considered if required (Emde et al., 2010). Sophisticated variance reduction methods have been implemented (Buras and Mayer, 2011) which enable the calculation of radiances for scattering media characterized by strongly peaked phase functions without any approximations. The MYSTIC model has been validated in various model intercomparison studies (Cahalan et al., 2005; Kokhanovsky et al., 2010a; Emde et al., 2015, 2018; Korkin et al., 2020; Zawada et al., 2021) and by comparison to benchmark results, and always agreed well to other participating radiative transfer codes.

Radiative transfer simulations for inhomogeneous scenes including clouds require a three-dimensional model including horizontal photon transport from cloudy parts into clear regions and vice versa. The Monte Carlo approach is well established for this type of simulations. However, for the simulation of spectra, the inherent statistical noise of the Monte Carlo simulations can be problematic, in particular when absorption features in the simulated spectra are very weak such as the characteristic

$NO_2$ absorption features in the spectral range from 400 to 500 nm. Running a Monte Carlo simulation for each wavelength sequentially would require an enourmeous amount of computational time, because the result for each wavelength would have its own statistical error required to be smaller than the weak absorption signal. The Absorption Lines Importance Sampling (ALIS) method (Emde et al., 2011) solves this problem. This method allows one to calculate the full spectrum based on photon path distributions sampled at a single wavelength. In order to take into account the spectral dependence of the absorption coefficient

a spectral absorption weight is calculated for each photon path. Further, at each scattering event the local estimate method (Marshak and Davis, 2005) is combined with an importance sampling method to take into account the spectral dependence of the scattering coefficient. Since each wavelength grid point is computed using the same photon path distribution, the statistical error of such a simulation is is almost independent of wavelength, i.e. it corresponds to a small offset of the complete spectrum. For DOAS type retrievals this error is completely removed by the polynomial fit to compute the differential optical thickness.

This statistical error decreases with the number of photons used in the simulation and converges towards the correct spectrum. The method it is very well suited to efficiently simulate radiance spectra in high-spectral resolution.

In the UV and visible spectral ranges, the standard retrieval algorithm is based on the DOAS technique (Platt, 2017): in a first step, the slant column density (SCD) is retrieved by spectral fitting of the observed solar spectra to absorption cross sections of trace gases. The SCD corresponds to the amount of trace gas along the average photon path from the Sun through

the atmosphere to the satellite sensor. In order to convert SCD into a vertical column density (VCD), the so-called air-mass factor is required, which is defined as the ratio between SCD and VCD. In clean regions, the retrieval error is dominated by the spectral fitting, while for polluted or cloudy regions, the uncertainty of the AMF becomes the dominant error source. The AMF is calculated using radiative transfer models.

MYSTIC includes the option to simulate 1D layer-AMFs or 3D box-AMFs (Schwaerzel et al., 2020). The concept of

layer/box-AMFs assumes that the trace gas concentration is small compared to the concentration of other gases, meaning that interaction of photons with trace gas molecules does not alter the photon path distribution in the atmosphere. Layer-AMFs are calculated from the photon path length distribution in each individual altitude layer of the model atmosphere as described in Deutschmann et al. (2011). MYSTIC calculates layer-AMFs for 1D plane-parallel or spherical atmospheres, and also for 3D model atmospheres. In the latter case the photon pathlengths are integrated horizontally over the full domain. Note that these

"3D" layer-AMFs still include the impact of 3D cloud scattering. In DOAS type retrievals the layer-AMFs are used together with the a priori $NO_2$ altitude profile to compute the total AMF:

$$\text{AMF} = \frac{\sum_l \text{AMF}_l \cdot x_l}{\sum_l x_l} \tag{1}$$

Here $l$ is the layer index, $\text{AMF}_l$ the layer-AMF and $x_l$ the partial column density (of NO$_2$) for layer $l$. This AMF is then used to convert from slant column density (SCD) to vertical column density (VCD):

$$\text{VCD} = \text{SCD}/\text{AMF} \qquad\qquad (2)$$

Note that in the literature, layer-AMFs are commonly called box-AMF (e.g. Deutschmann et al. (2011)), which is a confusing terminology, because they do not refer to model grid boxes. MYSTIC also enables the calculation of real "box"-AMFs which are derived from the 3D photon pathlength distribution, i.e. from the photon pathlengths in each 3D model grid cell. Box-AMFs are useful if one knows a 3D a priori NO$_2$ concentration distribution which can be used in the retrieval to convert from SCD to VCD (Schwaerzel et al., 2020). All currently available operational retrieval algorithms apply 1D a priori altitude concentration profiles, therefore they can not use box-AMFs.

Using MYSTIC, we may study how the layer-AMFs are modified by scattering from clouds in the neighborhood. Comparing the layer-AMFs of a clear sky atmosphere with the layer-AMFs influenced by clouds, we may estimate the retrieval error of, e.g., NO$_2$ vertical column densities (VCDs). Working with simulated layer-AMFs allows us also to study the impact of the vertical NO$_2$ concentration profile on the retrieval error. Since the influence of trace gases on the photon pathlength distribution and thus on layer-AMF is negligible, we may use the layer-AMFs of one radiative transfer simulation to estimate the error for various assumed NO$_2$ concentration profiles. Such an analysis is presented in part II of this publication series (Yu et al., 2021). For this reason it is not necessary to include simulations for different NO$_2$ profiles in the synthetic dataset.

## 3 2D box-cloud scenario

### 3.1 Definition of Setup

#### 3.1.1 Molecular atmosphere

The molecular atmosphere was defined according to the midlatude summer standard atmosphere of Anderson et al. (1986). The NO$_2$ number concentration was modified corresponding to the highly polluted case (see Fig. 1) with a vertical number density of $1.6 \cdot 10^{16}$ molec/cm$^2$, with most of the NO$_2$ located within the atmospheric boundary layer. As mentioned before we may use the layer-AMFs to investigate the impact of cloud scattering on the trace gas concentration retrieval. The layer-AMFs are independent of the trace gas profiles, for this reason we define only one NO$_2$-profile, but we can investigate retrieval errors also for different profiles including non-polluted cases (see also Yu et al. (2021)). We have chosen a fine vertical resolution of the model atmosphere in the lower part of the atmosphere, between 0 km and 12 km altitude the layer thickness is about 150 m. The vertical resolution from 12–25 km is 1 km, from 25–50 km 2.5 km and from 50–100 km 5 km. We have chosen the fine vertical resolution in the lower part of the atmosphere in order to resolve the vertical dependency of layer-AMF in the region of interest.

Rayleigh scattering cross sections were calculated using the parameterization by Bodhaine et al. (1999). For the visible spectral range from 400–500 nm we used the following absorption cross sections: NO$_2$ by Vandaele et al. (1998), O$_3$ by

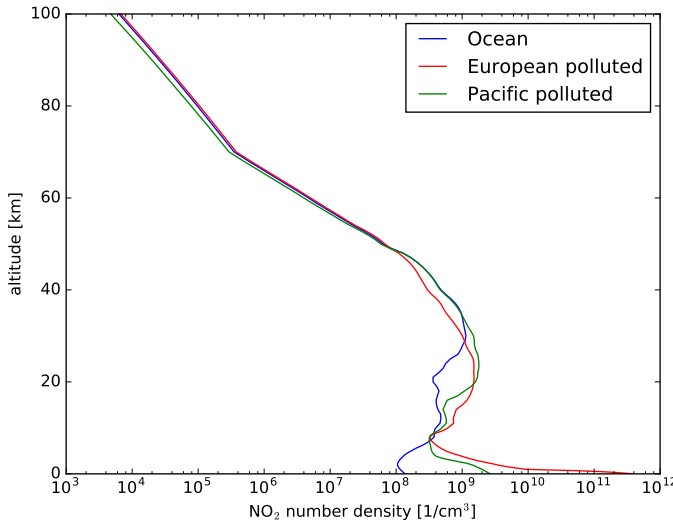

**Figure 1.** NO$_2$ number concentration profiles for typical scenarios taken from the CAMELOT study (Levelt et al., 2009).

Serdyuchenko et al. (2014), and O$_4$ by Thalman and Volkamer (2013). Absorption cross sections for the O$_2$A-band region were calculated using the line-by-line model ARTS (Eriksson et al., 2011) with line parameters from the HITRAN2012 molecular spectroscopic database (Rothman et al., 2013).

### 3.1.2 Box-cloud definition

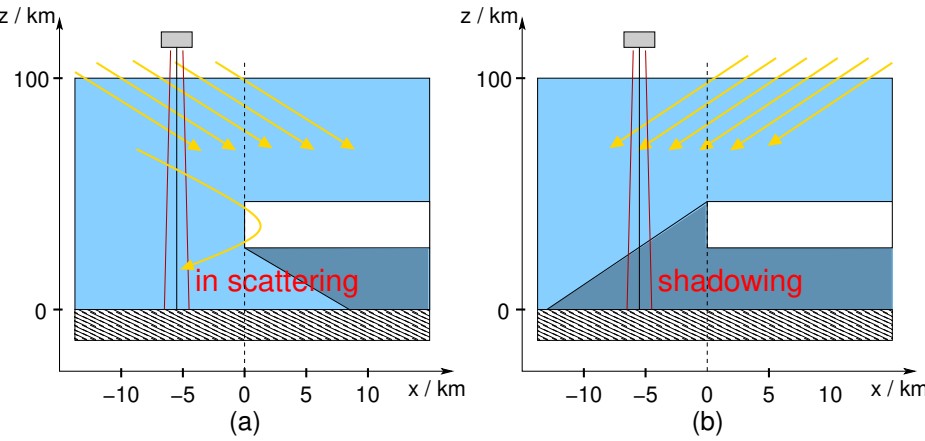

**Figure 2.** Schematic of 2d box-cloud scenario: (a) shows the in-scattering geometry and (b) the shadowing geometry.

5    In order to investigate the effect of in-scattering or shadowing in clear regions near clouds we defined a simple setup including a 2D-cloud, as shown schematically in Fig. 2. MYSTIC employs periodic boundary conditions, therefore, in order to avoid the effects of multiple cloud edges, we defined a large 2D domain extending from 0–200 km in the $x$-direction. Half of the

|  | liquid water cloud | ice water cloud |
|---|---|---|
| cloud optical thickness | 1, 2, 5, **10**, 20 | 1, 2, **5**, 10, 20 |
| cloud bottom height [km] | **2**, 5, 10 | 5, **9**, 12 |
| effective radius [$\mu$m] | 10 | 30 |
| optical properties | Mie | Baum (V3.6) |
| cloud geometrical thickness [km] | 0.2, **1**, 2, 4, 8 | |
| surface albedo | 0.02, **0.05**, 0.1, 0.15, 0.2, 0.3 | |
| solar zenith angle [°] | 20, 30, 40, **50**, 60, 70, 80 | |

**Table 1.** Settings for the box cloud simulations. Bold numbers indicate the base cases. For surface albedo, solar zenith angles and cloud geometrical thickness, the same settings were used for liquid and ice water clouds.

domain (0–100 km) was cloud-free and the other half includes the homogeneous box cloud (100–200 km). Since MYSTIC uses periodic boundary conditions, this means that next to the cloudy region, there is again a 100 km wide clear region. In the $y$-direction the cloud is extended to infinity.

We defined two types of box cloud base cases, corresponding to liquid and ice water clouds respectively: For the liquid water cloud the base height was set to 2 km and the top height to 3 km. The cloud droplet effective radius was set to 10 $\mu$m, a typical value for liquid water clouds. The cloud droplet radii distribution follows a gamma distribution with an effective variance of 0.1. The corresponding optical properties were calculated using the Mie tool provided with libRadtran (Wiscombe, 1980). The cloud optical thickness was set to 10.

For the ice cloud the altitude was set to 9–10 km. The optical properties of ice crystals were taken from the database by Yang et al. (2013) and Baum et al. (2014) assuming a general habit mixture. The effective radius was set to a typical value of 30 $\mu$m and the cloud optical thickness was set to 5, a relatively large value for which we expect significant 3D scattering effects.

Reflectance spectra were simulated for an imaginary nadir viewing sensor with a 1x1km$^2$ field-of-view for pixels along a line starting at a distance of 15 km away from the cloud egde in the clear region and ending at a distance of 10 km in the cloudy region. For the base cases, the solar zenith angle is set to 50° and the surface albedo to 0.05.

Aerosols were not included, although aerosol scattering also has a significant impact on the NO$_2$ retrieval. However, in this study, we aim to quantify the impact on cloud scattering on the retrieval. When both, aerosols and clouds are included, it becomes difficult to disentangle the impacts of cloud and aerosol scattering. Therefore, we decided to include only clouds.

Starting from the base case, we varied the following parameters: cloud optical thickness, cloud bottom height, cloud geometrical thickness, solar zenith angle, and surface albedo. The simple setup allows us to study the sensitivity of the NO$_2$ VCD retrieval error on these parameters separately. The settings for all parameters are summarized in Table 1.

For all combinations of parameters we also calculated radiance spectra for a corresponding 1D cloud layer setup, where the cloud is extended horizontally over the full model domain. The cloud optical and microphysical properties are exactly the same as for the 3D cloud simulations. Further we calculated the corresponding clearsky spectra for all solar zenith angles and surface

albedos using the same background atmosphere as for the simulations with clouds. The difference between the 3D simulations and the 1D simulations corresponds to the impact of horizontal photon scattering.

### 3.1.3 Monte Carlo simulation settings

The variance reduction methods VROOM (Buras and Mayer, 2011), which reduce the statistical noise in Monte Carlo radiative transfer simulations including cloud scattering, were enabled in all simulations except for the completely clear-sky domain. For all cases (1D clear-sky, 1D cloud layer, and 2D box cloud) we have performed 100 times the same simulations with $10^5$ photons per pixel. From the 100 results we calculated the standard deviations for reflectances and layer-AMFs. Radiances were simulated in the spectral range from 400–500 nm with 0.2 nm resolution (VIS), and from 755–775 nm with 0.005 nm spectral resolution ($O_2$A-band). Moreover layer-AMFs were calculated at 460 nm. The fine spectral resolution in the $O_2$A-band was chosen to resolve the individual spectral lines. The ALIS method (Emde et al., 2011) was applied for all simulations.

## 3.2 Synthetic data for box-cloud and clear-sky

### 3.2.1 Simulated spectra

Fig. 3 shows the reflectance as a function of distance from the cloud edge for three selected wavelengths of the dataset (400 nm, 500 nm and 760 nm). All results correspond to the two base cases, i.e. the solar zenith angle is $50°$ and the surface albedo 0.05. The region from -15 km to 0 km corresponds to clear sky pixels and the region from 0 km to 10 km corresponds to cloudy pixels. The solid lines show the 3D reflectance simulations and the dashed lines the 1D clear sky and cloud reflectance simulations. In the in-scattering region (left panels), the cloudy reflectance is larger than the clear sky reflectance. The right panels clearly show the cloud shadow near the the cloud edge, which has a much larger extent for the higher cloud. This can easily be explained geometrically.

Fig. 4 shows example spectra for the liquid water cloud at low altitude. The top panel (a) shows reflectance spectra in the range from 400–500 nm. The black line corresponds to a simulation for clear sky without any clouds in the domain. The blue line corresponds to a spectrum in the cloud shadow region (center of the square pixel is 1.5 km away from the cloud edge) and the green line corresponds to a pixel centered 10.5 km away from the cloud edge. In the shadow, the reflectance spectrum is lower than the clear-sky spectrum, as expected. Far away from the cloud edge, the reflectance spectrum is higher than the clear-sky spectrum, because the cloud scatters radiation from the cloudy part of the domain towards the clear part.

All spectra show the characteristic absorption features, which can much better be recognized in the differential optical thickness $D$ obtained by substracting a third degree least square polynomial fit to the logarithm of the reflectance spectrum $P_3$:

$$D(\lambda) = \ln(I(\lambda)) - P_3(\lambda) \tag{3}$$

$D(\lambda)$, panel (b) Fig. 4, shows the characteristic $NO_2$ absorption features and the $O_2$-$O_2$-absorption band around 480 nm which can be used to retrieve information about the cloud height. Note that $D(\lambda)$ is a smooth function although simulated using a

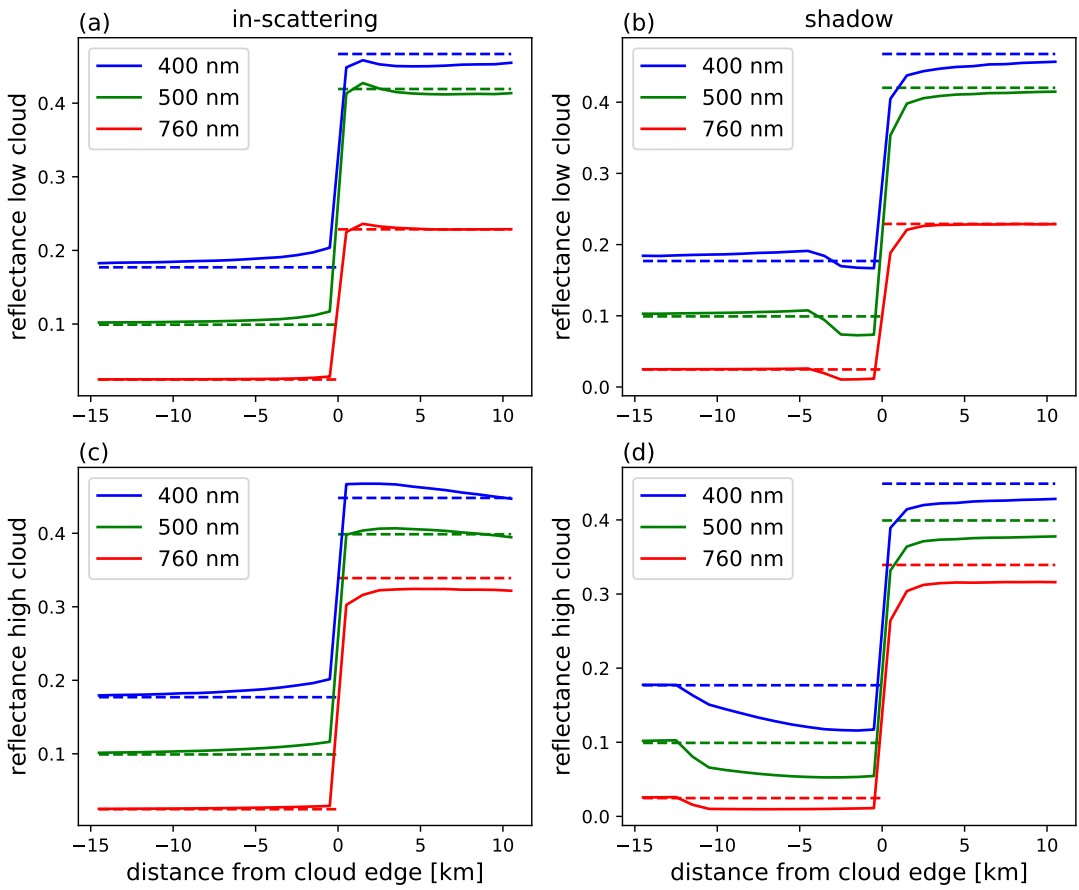

**Figure 3.** Reflectance as a function of distance from the cloud edge for the base case for selected wavelengths included in dataset: (a) in-scattering for liquid water cloud at low altitude, (b) shadowing for liquid water cloud at low altitude, (c) in-scattering for liquid water cloud at high altitude, (d) shadowing for liquid water cloud at high altitude. Solid lines correspond to 3D and dashed lines to 1D simulations.

Monte Carlo model. As mentioned above, this is possible with the ALIS method, which samples the full spectrum based on the same photon path distribution. Each of the reflectance spectra has a small statistical bias (smaller than 1% for the number of photons used in the simulation). This bias is completely removed by substraction of the polynomial fit. $D$ for the clear-sky simulation is very similar for $D$ for the box-cloud simulation for the pixel centered at 10.5 km away from cloud edge. However, in the cloud shadow region, there are obvious differences.

Panel (c) in Fig. 4 shows spectra in the $O_2$A-band region for the same pixels. These are computed in very high spectral resolution to resolve the individual spectral lines. Again we find much smaller reflectance values compared to clear-sky for the cloud shadow pixel and slightly larger values for the pixel further away from the cloud edge.

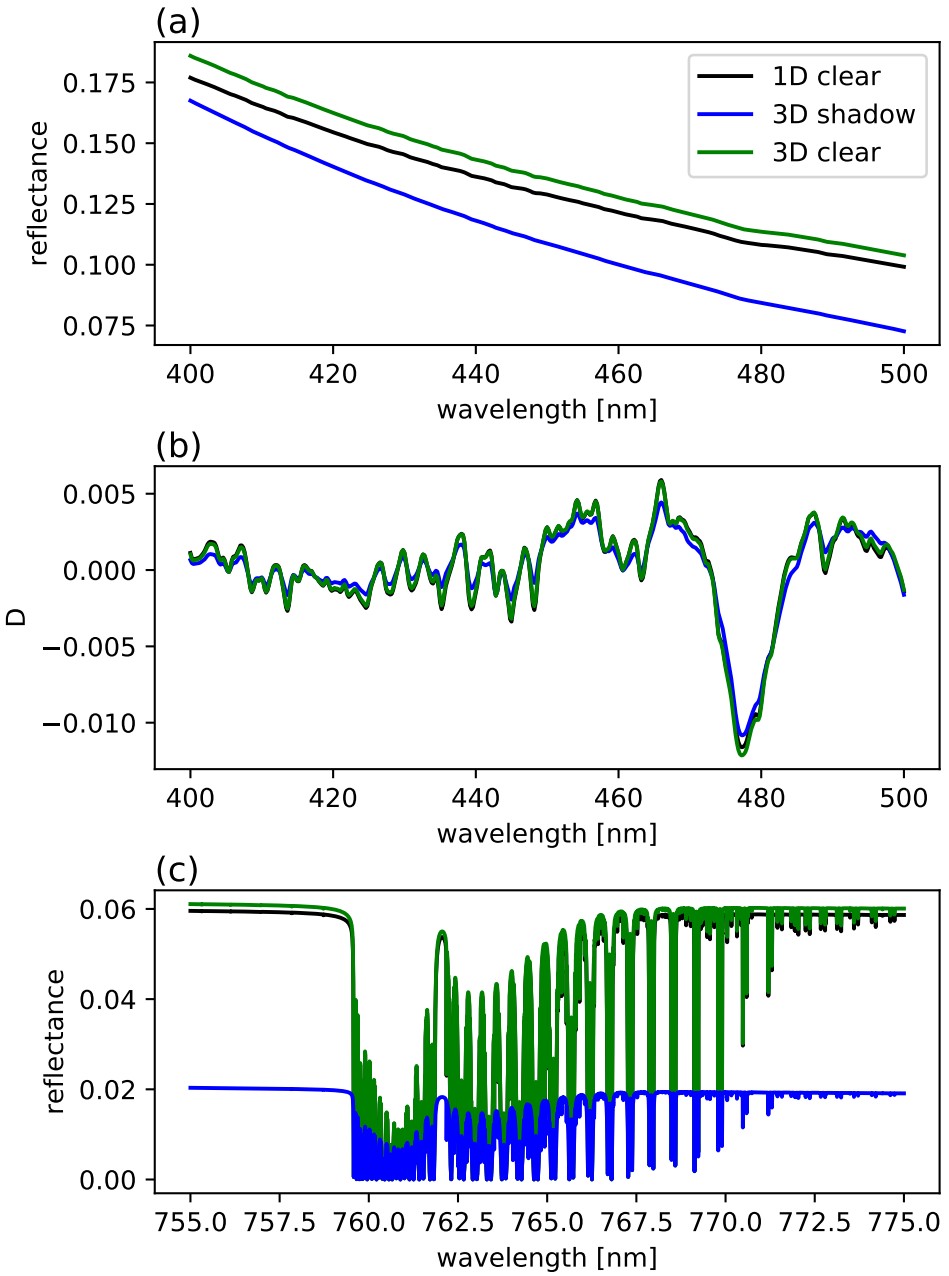

**Figure 4.** (a) Reflectance spectra for the spectral range from 400–500 nm, (b) differential optical thicknesses derived from these spectra. (c) corresponding reflectance spectra in the $O_2$A-band region. The black lines correspond to a 1D clear sky simulation and the blue and green lines to 3D simulations. The blue lines are for a pixel centered 1.5 km away from the cloud edge in the cloud shadow, and the green lines are for a pixel centered 10.5 km away from cloud edge in the clear region.

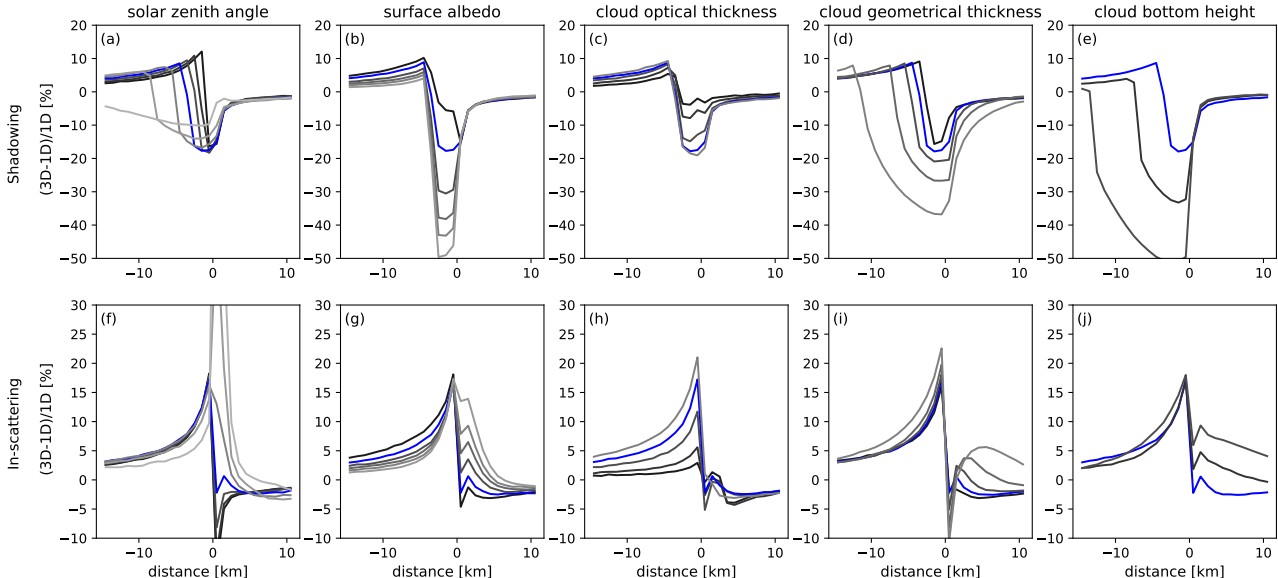

**Figure 5.** Relative difference between 3D and 1D reflectance at 460 nm for the liquid water cloud at 2–3 km altitude. Negative distances from the cloud edge correspond to pixels in the clear region and positive differences to pixels in the cloudy region of the domain. The individual columns are for the parameters defined in Table 1, see also the figure titles. The upper row (panels (a)–(e))) is for the cloud shadow and the lower row (panels (f)–(j)) for in-scattering by the cloud. The blue line in each figure corresponds to the base case and the other lines correspond to the parameter range as in Table 1, starting from the lowest value in black to the highest value in gray.

### 3.2.2 3D impact depending on solar zenith angle, surface albedo and various cloud parameters

Fig. 5 shows the impact of 3D cloud scattering on the reflectance at 460 nm depending on the parameters defined in Table 1. All panels show relative differences:

$$\Delta I_{\mathrm{rel}}(\lambda) = \frac{I_{3D}(\lambda) - I_{1D}(\lambda)}{I_{1D}(\lambda)} \tag{4}$$

5  $I_{3D}(\lambda)$ are reflectance spectra obtained by 3D radiative transfer simulations. $I_{1D}(\lambda)$ are corresponding reflectance spectra obtained from 1D simulations, where clear-sky simulations are taken for clear pixels (negative $x$-values) and 1D-cloud simulations for cloudy pixels (positive $x$-values).

Panel (a) shows that with increasing solar zenith angle from 20° (black line) to 80° (light grey line) the horizontal extension of the cloud shadow indicated by negative $\Delta I_{\mathrm{rel}}$, also increases. The relative difference $\Delta I_{\mathrm{rel}}$ in the cloud shadow is larger
10  than -15% for SZA $\leq$60°. Next to the cloud shadow, in the clear region corresponding to negative $x$-values, $\Delta I_{\mathrm{rel}}$ is positive because photons are scattered from the cloudy part of the domain into the clear part. Generally, for cloudy pixels corresponding to positive $x$-values, $\Delta I_{\mathrm{rel}}$ is negative and does not depend significantly on SZA. For in-scattering (panel (f)), $\Delta I_{\mathrm{rel}}$ is positive due to cloud scattering near the cloud edge. The amount is up to 20% at the cloud edge and it continually decreases with distance from the cloud edge. For the cloudy pixel at the edge of the cloud, $\Delta I_{\mathrm{rel}}$ is negative for SZA $\leq$50°  whereas it is

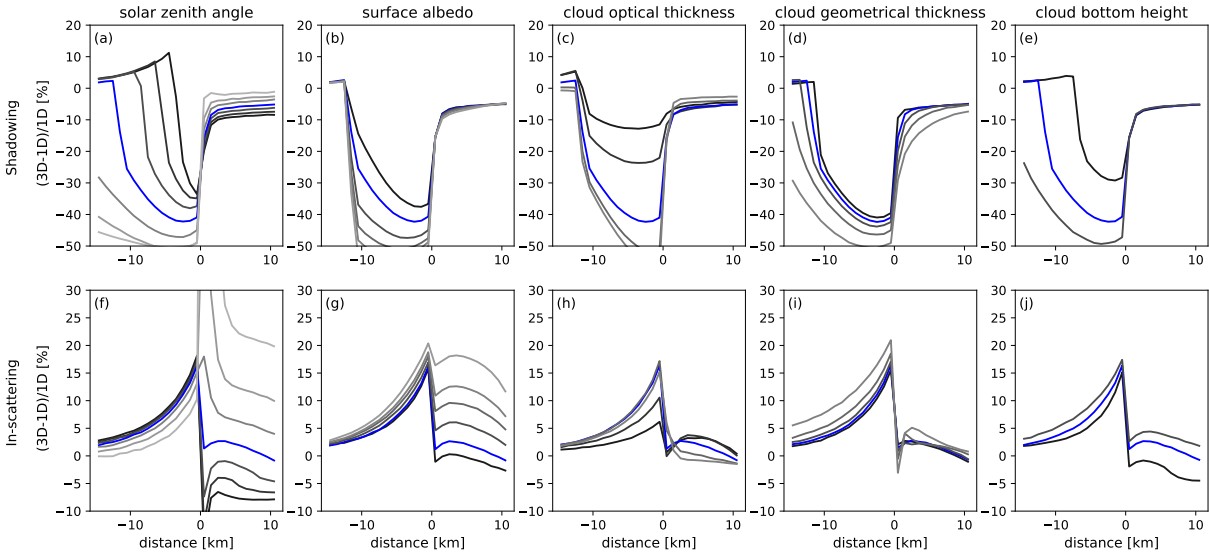

**Figure 6.** Same as Fig. 5 but for the ice water cloud at 9–10 km altitude. All settings are summarized in Table 1.

positive for SZA $>50°$. The explanation for this dependency is that for slant incidence more photons enter the cloud from the side and are scattered upwards towards the sensor.

Panels (b) and (g) show the dependencies on surface albedo: The relative difference $\Delta I_{\rm rel}$ in the cloud shadow increases with increasing surface albedo, up to 50% for a surface albedo of 0.3 (light gray line). The in-scattering in the clear region ($x <0$ km)

shows the opposite behaviour: as the albedo increases $\Delta I_{\rm rel}$ decreases. For cloudy pixels ($x >0$ km), there is no dependency on surface albedo for the shadowing geometry (upper panel), whereas for in-scattering (lower panel), $\Delta I_{\rm rel}$ increases with increasing surface albedo.

Panels (c) and (h) show the dependence of $\Delta I_{\rm rel}$ on cloud optical thickness $\tau$: Both effects, enhancement by in-scattering and reduction by shadowing increase as the cloud optical thickness increases from 1 (black line) to 20 (light grey line).

The other panels show the dependencies of $\Delta I_{\rm rel}$ on cloud geometrical thickness (panels (d) and (i)) and cloud bottom height (panels (e) and (j)). The cloud shadow is highly sensitive to those two cloud parameters, which is easily understood by simple geometric considerations: As the cloud's geometrical thickness increases, the size of the shadow increases and the same is true for increasing cloud bottom height. The amount of in-scattering in the clear part of the domain ($x <0$ km) is similar for all cloud geometrical thicknesses and all cloud bottom heights, given a constant cloud optical thickness of 10.

Fig. 6 shows the same dependencies but for the ice cloud at 9-10 km altitude with smaller cloud optical thickness values (see Table 1). The dependencies of $\Delta I_{\rm rel}$ on the various paramerters are similar for the ice cloud as for the water cloud. However, $\Delta I_{\rm rel}$ in the cloud shadow is comparatively large (more than 50%), which is simply explained by the higher altitude of the cloud producing a larger shadow area. Enhancement by in-scattering is also increased compared to the liquid water cloud.

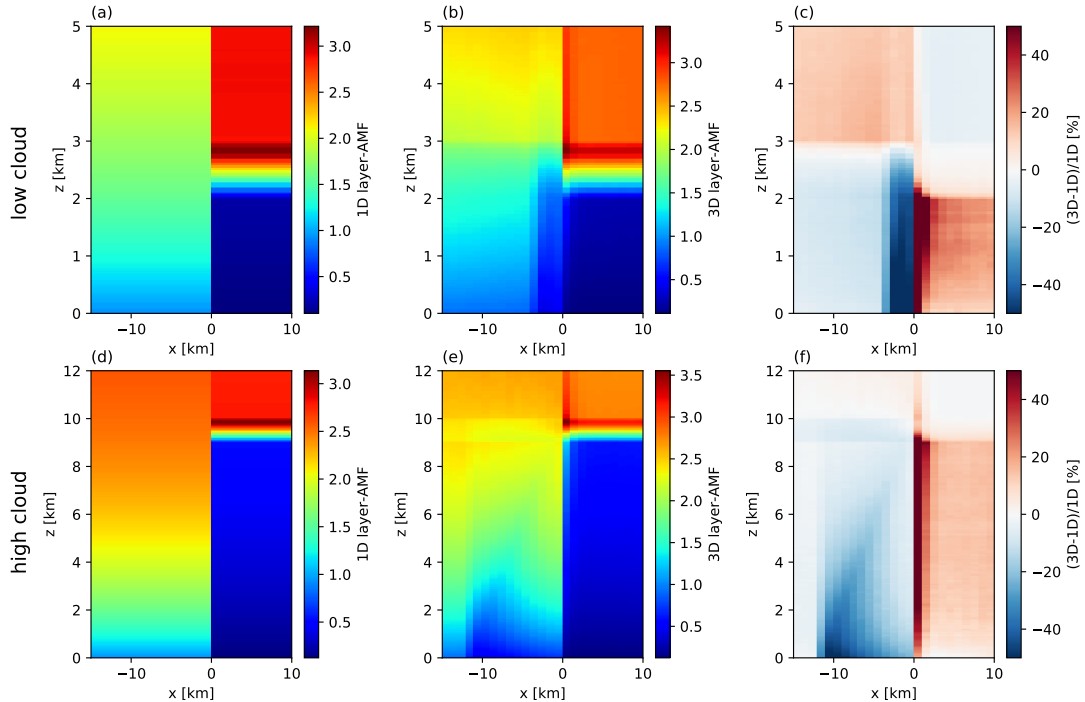

**Figure 7.** Layer-air-mass-factors for the two base cases (SZA=50°, surface albedo=0.05) for cloud shadow geometry: layer-AMFs calculated using 1D approximation (clear-sky from $x$=-15 km to $x$=0 km and 1D cloud layer from $x$=0 km to $x$=10 km) for low cloud (a) and high cloud (d), 3D simulation of layer-AMFs for low cloud (b) and high cloud (e), relative difference between 1D and 3D simulation for low cloud (c) and high cloud (f). Note that the altitude range $z$ in the upper panels (a)–(c) is from 0 km to 5 km and in the lower panels (d)–(f) from 0 km to 12 km.

### 3.2.3 Layer airmass factors

For all settings listed in Table 1 the dataset includes corresponding layer-AMFs. Fig. 7 presents layer-AMF as a function of height and distance from cloud edge. The left panels show 1D simulations for a clear atmosphere (-15 km<x<0 km) and for the same atmosphere with 1D cloud layer included (0 km<x<10 km). The upper panels (a)–(c) are for the liquid water cloud

5    at 2–3 km altitude and the lower panels (d)–(f) for the ice water cloud at 9–10 km altitude. The layer-AMF in clear-sky at high altitudes is approximately 2.5, corresponding to the geometrical photon pathlength through a layer, normalized to the thickness of the layer. Towards lower altitudes the photons experience Rayleigh scattering, therefore not all photons reach the surface and the mean photon path length decreases. Within the cloud layer the photon pathlength and thus the layer-AMF is increased due to multiple scattering in the cloud. Also above the clouds, layer-AMF values are larger than corresponding clear-sky values.

10   Below the cloud, the layer-AMF decreases towards very small values, because only very few photons penetrate through the cloud twice. Most of the photons that reach below the cloud layer are trapped and finally absorbed by the surface or by gas molecules.

The middle panels (b) and (e) show the 3D simulations of layer-AMF for the domain including the box-cloud and the right panels (c) and (f) show the relative difference between 3D and 1D simulations. In the cloud shadow region, the layer-AMF is reduced compared to clear-sky. Below the cloud, it increases because photons which are in case of a 1D cloud layer trapped below the layer can escape in the clear part of the domain and then be scattered towards the sensor. The maximum relative difference in the cloud shadow region is about 50%.

The relative difference between the 1D and 3D layer-AMF is directly related to the impact of 3D cloud scattering on the retrieval error of, e.g., $NO_2$ number concentration profiles. $NO_2$ retrieval algorithms calculate layer-AMFs using a 1D radiative transfer model and use these to convert from absorption line strength to number concentration (the absorption optical thickness of a certain layer is the product of layer-AMF, absorption cross section and number concentration). Therefore, the relative difference between 1D and 3D simulation of layer-AMFs directly corresponds to the retrieval error of $NO_2$ in a certain layer. Depending on the vertical profile of $NO_2$ the retrieval error of the vertical column density will be more or less affected.

A detailed sensitivity study of the $NO_2$ retrieval error based on the box-cloud synthetic dataset is presented in Yu et al. (2021). Largest retrieval biases were found in the cloud shadow region, typically the errors are in the range of 10–100% for the polluted scenario. The bias increases with solar zenith angle, decreases with surface albedo and it increases with cloud optical thickness. The dependency on cloud geometrical thickness and cloud bottom height is less pronounced. Yu et al. (2021) also show that the cloud effects are much stronger for polluted cases compared to non-polluted cases, the maximum retrieval bias for the polluted profile is 95% for the base case settings and for the clean profile it is reduced to 6%. Various different $NO_2$ profile shapes have been investigated in addition, clearly demonstrating that the retrieval bias depends on the altitude where most of the $NO_2$ is located. The synthetic data was also applied to investigate the dependancy of the retrieval bias on the spatial resolution of the instrument. The synthetic data is created for a sensor footprint of $1 \times 1\,km^2$. By averaging, spatial resolutions between 3-15 km could be investigated. As expected, the retrieval bias decreases with increasing spatial resolution due to spatial averaging. The cloud shadow effect strongly depends on the cloud shadow fraction in a pixel.

## 4 Synthetic dataset for realistic cloud scene

In this section we present the synthetic dataset including realistic 3D clouds and observation geometries for LEO (Low Earth Orbit) and GEO (Geostatinary Earth Orbit) satellites. As in the previous section the data includes layer-AMFs, and reflectance spectra in the visible range from 400 to 500 nm and in the $O_2A$-band.

### 4.1 Description of cloudy scene

#### 4.1.1 ICON LES simulation of 3D liquid and ice cloud fields

As model input we selected a cloud scene from the Large Eddy simulation (LES) model ICON (Dipankar et al., 2015; Zängl et al., 2015; Heinze et al., 2017) simulated for the $29^{th}$ July 2014 at 12:00 UTC, which has been generated within the HDCP[2] project (www.hdcp2.eu). The scene includes Germany, the Netherlands, parts of surrounding countries and part of the North

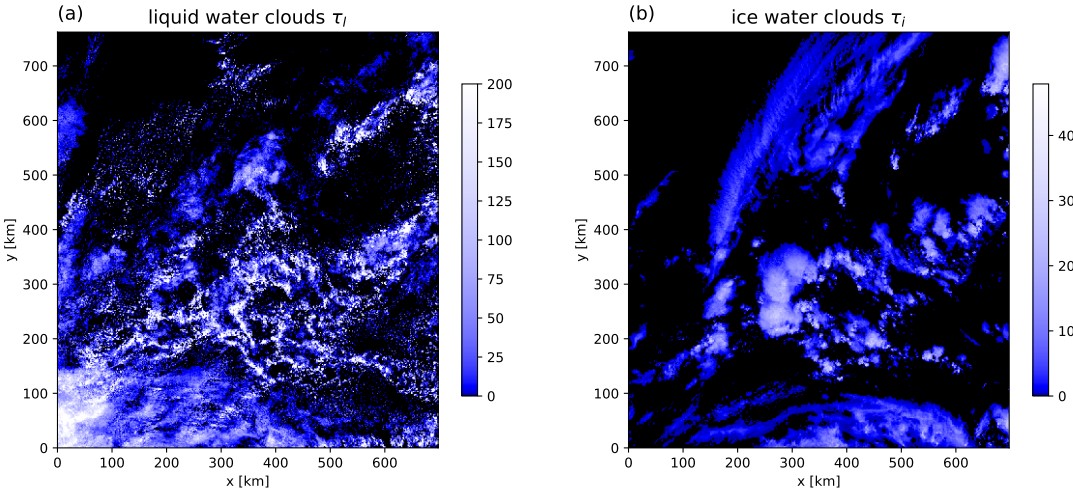

**Figure 8.** Vertically integrated cloud optical thickness for the simulated cloud scene from the ICON LES model ((a) liquid water cloud (b) ice water clouds).

sea. The ICON model has been validated against satellite based observational data (Heinze et al., 2017), and a good statistical agreement between observations (active and passive) and the model clouds was found, therefore we may assume that the ICON generated clouds are realistic. The spatial resolution of the clouds in ICON is approximately $1.2 \times 1.2\,\mathrm{km}^2$ in the horizontal and better than $100\,\mathrm{m}$ in the vertical. The simulation includes all cloud types that are typical for Europe, i.e. shallow cumulus, cirrus, stratus and convective clouds.

The ICON model provides 3D fields of cloud liquid and ice water content which were transformed to extinction coefficient fields, that are used as input for the radiative transfer simulations. This transformation requires assumptions about the effective radii of cloud droplets and crystals. We followed the approach based on physical parameterizations as outlined in Bugliaro et al. (2011).

Fig. 8 shows the vertically integrated optical thickness of liquid water clouds (a) and ice water clouds (b). Here we can already identify some typical cloud types: e.g. shallow cumulus fields consisting of small scattered liquid water clouds in the upper left part of the scene (around $x$=150 km, $y$=550 km), a thunderstorm with a large convective cloud cell (around $x$=280 km, $y$=300 km) with a typical cirrus shield on top. The scene includes multi-layer clouds and also mixed-phase clouds.

### 4.1.2 MYSTIC reflectance simulation for 3D cloud scene

In order to compare the ICON scene with real satellite data to check whether the model clouds are realistic, we simulated a reflectance image using the spectral response function from Sentinel3-SLSTR band 1, with a central wavelength of 554 nm and a bandwidth of 19.26 nm. The simulation was done for nadir observation geometry on a spatial resolution of $1.2 \times 1.2\,\mathrm{km}^2$ corresponding to $588 \times 624$ pixels for the full domain. The solar zenith angle was set to $30°$ and the solar azimuth angle to $13°$. The surface albedo map was taken from MODIS (Schaaf et al., 2002). Gaseous components were defined according to the US

standard atmosphere from Anderson et al. (1986). The optical properties of liquid droplets were calculated using Mie theory, and those of ice crystals were derived from the data for a general habit mixture from Yang et al. (2013) and Baum et al. (2014).

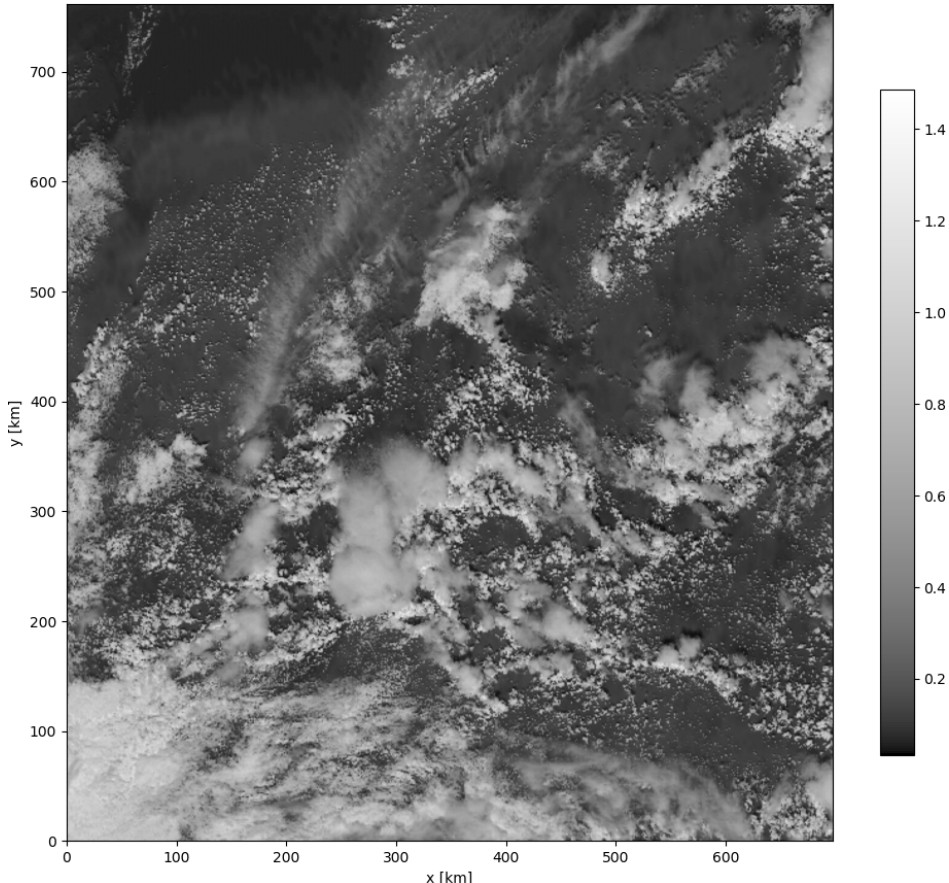

**Figure 9.** Reflectance simulation for Sentinel3-SLSTR, band 1 centered at 555 nm.

The resulting synthetic satellite image is shown in Fig. 9. The image looks very similar to a real satellite image, including 3D cloud structures and cloud shadows. From this synthetic image we calculated several metrics to quantify cloud features (i.e.
5  cloud geometric and radiance fractions, cloud shadow fraction and H-metric) and compared the results to metrics derived from VIIRS satellite images. For the calculation of the H-metric of a $7 \times 7 \, \text{km}^2$ area corresponding to the size of a TROPOMI pixel we take into account 36 simulated band reflectances contained in this area. The H-metric is the standard deviation of these reflectances divided by their mean value and it provides an estimate of the variation of reflectance within a TROPOMI pixel (Kylling et al., 2021). We found that the metrics derived from the synthetic data are comparable to the ones derived from the
10  real data (more details are given in Appendix A). This scene will be used in the following to generate synthetic Sentinel-5 data for validation of $NO_2$ retrieval algorithms.

## 4.2  Setup of radiative transfer model

### 4.2.1  Molecular absorption

The molecular atmosphere was set up according to Section 3.1.1, i.e. midlatitude standard atmosphere with modified $NO_2$ profile, constant over the full domain. We have chosen a constant $NO_2$ profile because we aim to investigate the impact of realistic clouds on the retrieval results. When we include an inhomogeneous $NO_2$ profile it is not easily possible to quantify this impact, e.g. to figure out which type of clouds have the largest impact on the retrieval error. This is only possible when we have the same atmospheric background conditions over the full domain. For the visible spectral range the same absorption coefficients as in Section 3.1.1 were applied.

The FRESCO cloud retrieval algorithm (Wang et al., 2008) averages reflectances for three spectral bands within the $O_2A$-band: 758–759 nm (outside absorption band), 760–761 nm (center of absorption band), and 765–766 nm (wing of absorption band). Therefore, in order to reduce data storage requirements, only the values of the three averages for each simulated spectrum are stored in the synthetic data. For this purpose, the REPTRAN absorption parameterization (Gasteiger et al., 2014) ("fine" resolution) available in libRadtran, which can be used in combination with the ALIS method was found to be sufficiently accurate (reflectances agree to 3 digits after the decimal point) and has been applied instead of full line-by-line simulations.

### 4.2.2  Representative sun positions and satellite viewing angles

**Geostationary Orbit**

In order to obtain typical values for geostationary orbits the up-coming Ultra-violet Visible Near-infrared (UVN) instrument on Sentinel-4 was analyzed, and it was found that SZA varies between 20° and 90° for the area under study. Since the 3D radiative transfer model setup does not take into account the sphericity of the Earth the maximum SZA was set to 60°. For SZA<60° the solar azimuth angle SAA varies close to linearly between about 250°–360° and 0°–100°. The satellite viewing azimuth and zenith angles (VAA and VZA) are fixed for geostationary orbit. For the area under study VZA varies between 54° and 63° with a mean of 58.3°, and VAA varies between 163° and 175° with a mean of 196.3°.

**Low Earth Orbit**

In order to obtain representative angles for Low Earth Orbit, distributions of SZA, SAA, VZA, and VAA relevant for TROPOMI were calculated for the 14th of July, August, September and October for scenes covering the study region. It was found that SZA varies between 20° and 60° between individual scenes. Over a single scene it varies by about 11-13°. The SAA distribution is bimodal when SZA< 40°. The modes are centered around about 7° and 353° with a small spread of less than 10°. For larger SZA it has a single mode between 5° and 24°. VZA varies between 0° and 60° with a rather broad and flat distribution. The VAA distribution may be both unimodal and bimodal. In either case the modes are centred around about 110° and 282° with spread of a few degrees. Note that while for 1D radiative transfer simulations it is sufficient to only cover the relative solar and satellite azimuth angles, we need to take into account the absolute azimuth angles for 3D radiative transfer.

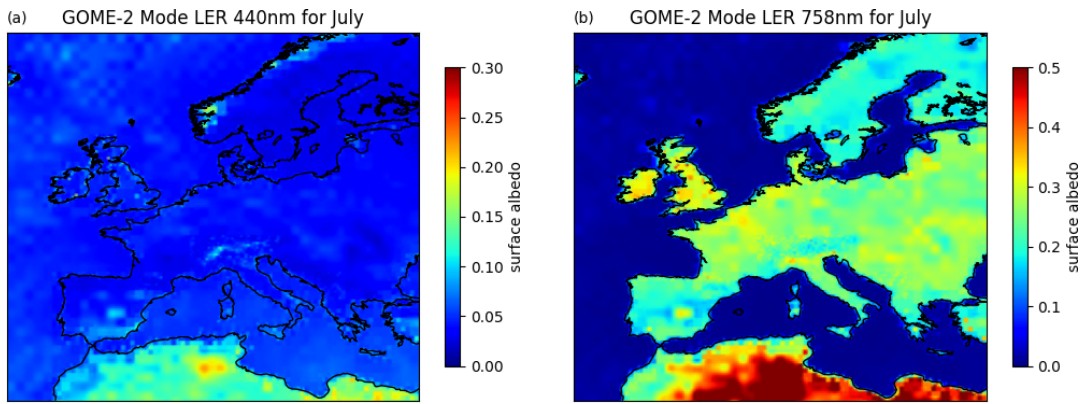

**Figure 10.** The GOME-2 July surface albedo map for 440 (a) and 758 nm (b). The data is from http://www.temis.nl/surface/albedo.html.

### 4.2.3 Representative surface albedos

The 3D simulations should cover a range of surface albedos representative for snow-free conditions for the study region. The surface albedo values for 440 nm and 758 nm as derived from GOME-2 are shown in Fig. 10. For visible wavelengths (440 nm) they are in the range between 0 and 0.2. At 757 nm (a wavelength close to the $O_2A$-band) the surface albedo increases to maximum values of about 0.5.

### 4.2.4 Example for visible band (400–500 nm)

In this section the synthetic data for a typical sun-observer geometry similar to the one shown in Section 4.1.2 is presented. Here, the solar zenith angle is 40°, the solar azimuth angle is 13°, the sensor viewing direction is nadir, and the surface albedo is 0.05.

The same quantities as for the 2D cloud scenario defined in Section 3.1.2 were simulated: reflectance spectra from 400–500 nm with a spectral resolution of 0.2 nm and layer AMFs at 460 nm. For the $O_2A$-band averaged reflectance values for the three bands as specified in Section 4.2.1 were calculated. The spatial resolution of the simulated sensor was set to approximately $7\times7$ km$^2$, which roughly corresponds to the spatial resolution of the TROPOMI instrument onboard Sentinel-5P and Sentinel-5. With this resolution we obtain $98\times104$ pixels for the full domain. Note that each simulated pixel includes 36 cloud pixels, consequently the simulations contain sub-pixel cloud inhomogeneity.

In order to obtain the statistical uncertainty of the Monte Carlo results, the simulation was repeated 100 times, each time running 1000 photons per pixel. From the 100 results the standard deviations for reflectances and layer-AMFs were calculated. Note that the number of photons was 100 times less than for the box-cloud and clear-sky cases presented in Section 3.2. Therefore, since the standard deviation of a Monte Carlo simulation is inversely proportional to the square root of the number of photons, the standard deviation of all simulations results (reflectance spectra and layer-AMFs) is increased by a factor of 10.

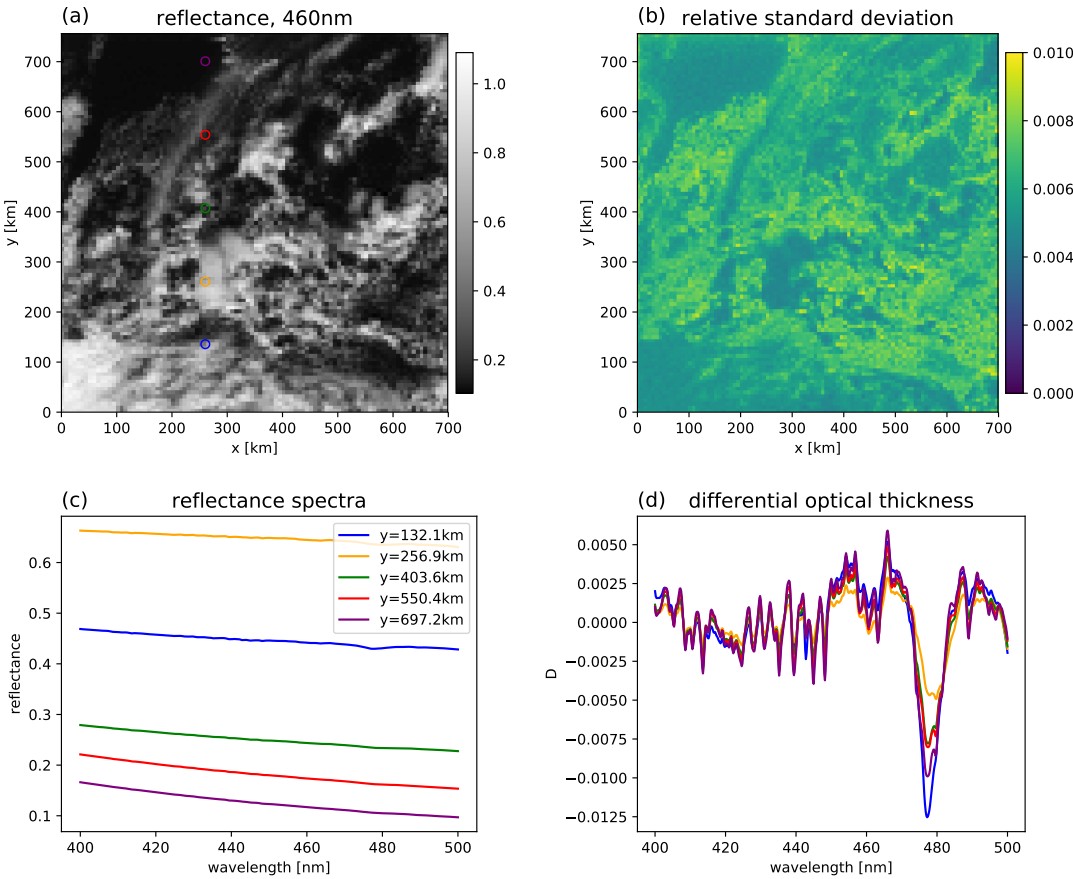

**Figure 11.** (a) Reflectance at 460 nm at a spatial resolution of about $7{\times}7\,\mathrm{km}^2$, (b) relative standard deviation of reflectance, (c) reflectance spectra for example pixels marked by circles in the reflectance image, (d) differential optical thickness calculated from the reflectance spectra.

Panel (a) in Fig. 11 shows the reflectance at 460 nm (spatial resolution approximately $7{\times}7\,\mathrm{km}^2$) and panel (b) shows the corresponding relative standard deviation which is generally smaller than 1%. The same cloud structures as in the high spatial resolution image (Fig. 9) are visible, but of course, several features as for instance the individual clouds in the shallow cumulus cloud field are not resolved. Panel (c) in Fig. 11 shows the reflectance spectra obtained for the pixels marked by coloured circles. The purple line corresponds to a clear pixel, which is marginally influenced from surrounding clouds. The red and the green pixels are situated between clouds and are obviously affected by them. The yellow pixel (highest reflectance) comprises the large convective cloud with cirrus shield on top and the blue pixel contains a stratocumulus cloud. Panel (d) in Fig. 11 shows the differential optical thicknesses derived from these spectra. The depth of the distinct $O_2$-$O_2$ absorption band around 480 nm decreases for the convective cloud including a cirrus at high altitude (yellow) and increases for the stratocumulus cloud at low altitude (blue) compared to clear sky (purple). The reason for this is that for the cirrus, the photon pathlength decreases compared to clear-sky because photons are reflected back to the sensor at a high altitude. For the stratocumulus, the

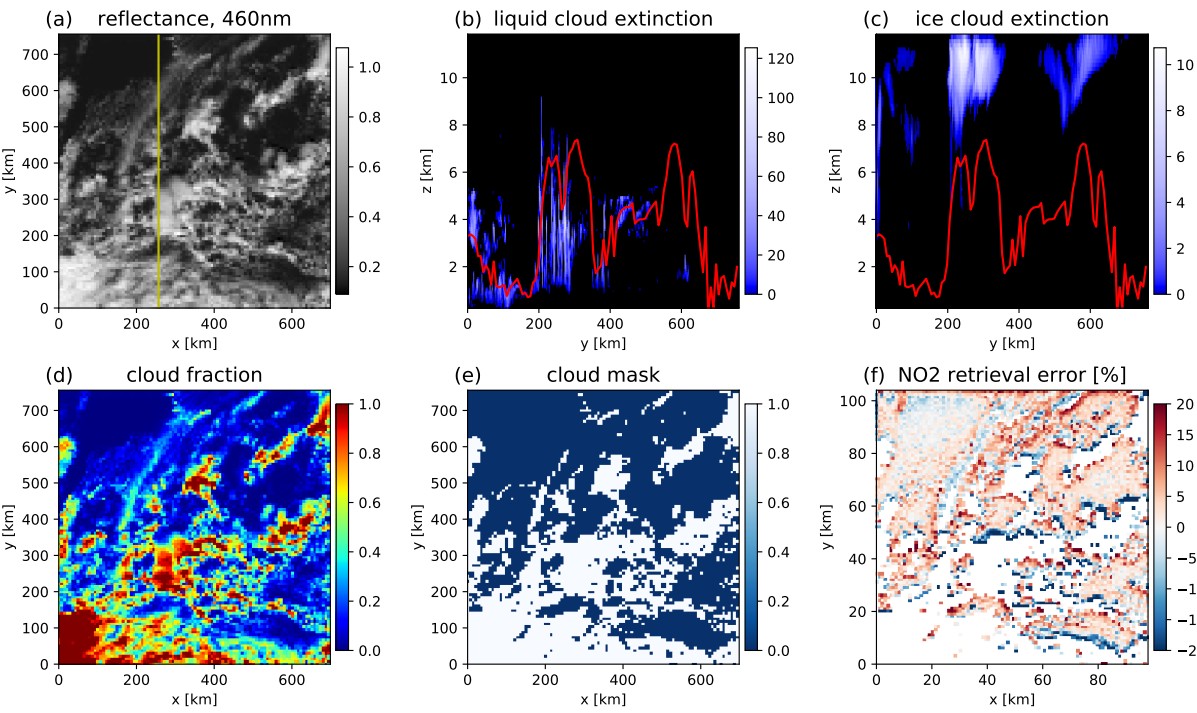

**Figure 12.** (a) Simulated reflectance at a spatial resolution of approximately $7\times7$ m$^2$. Vertical cross sections of extinction coefficients of liquid (b) and ice water clouds (c), along the yellow line drawn on the reflectance image. The red lines correspond to the effective cloud height derived from the $O_2$-$O_2$ absorption band. (d) Retrieved weighted radiometric cloud fraction, (e) cloud mask, all pixels with cloud fraction larger than 0.3 (white) are excluded from the $NO_2$ retrieval, (f) $NO_2$ retrieval error.

pathlength increases compared to clear-sky, because photons are multiple-scattered in the cloud-layer before they are reflected back to space. This figure clearly demonstrates that the $O_2$-$O_2$ absorption band can be used to retrieve information about cloud altitude.

Fig. 12 shows examplary results of an $NO_2$ retrieval algorithm (Blond et al., 2007; De Smedt et al., 2008) applied on the
5   synthetic data. Results for all cases and detailed analyses are presented in the accompanying publications by Yu et al. (2021) and Kylling et al. (2021). Panel (a) shows the same reflectance image as in Fig. 11. Panels (b) and (c) show extinction coefficients of liquid water and ice water clouds, respectively, as vertical cross sections along the yellow line drawn on the reflectance image. These figures show that liquid water clouds are generally more inhomogeneous than ice clouds. The extinction coefficients of liquid clouds are much larger than those of ice clouds (please note the different scales of the color bars). Liquid clouds are
10  present in the altitude range from a few hundred meters up to about 8 km, while ice clouds extend from about 2 km to 12.5 km (upper limit of the model domain). The red lines show the effective cloud height derived from the $O_2$-$O_2$-absorption band. Of course this does not correspond to the real geometrical cloud height: ice clouds are mostly above the effective cloud height

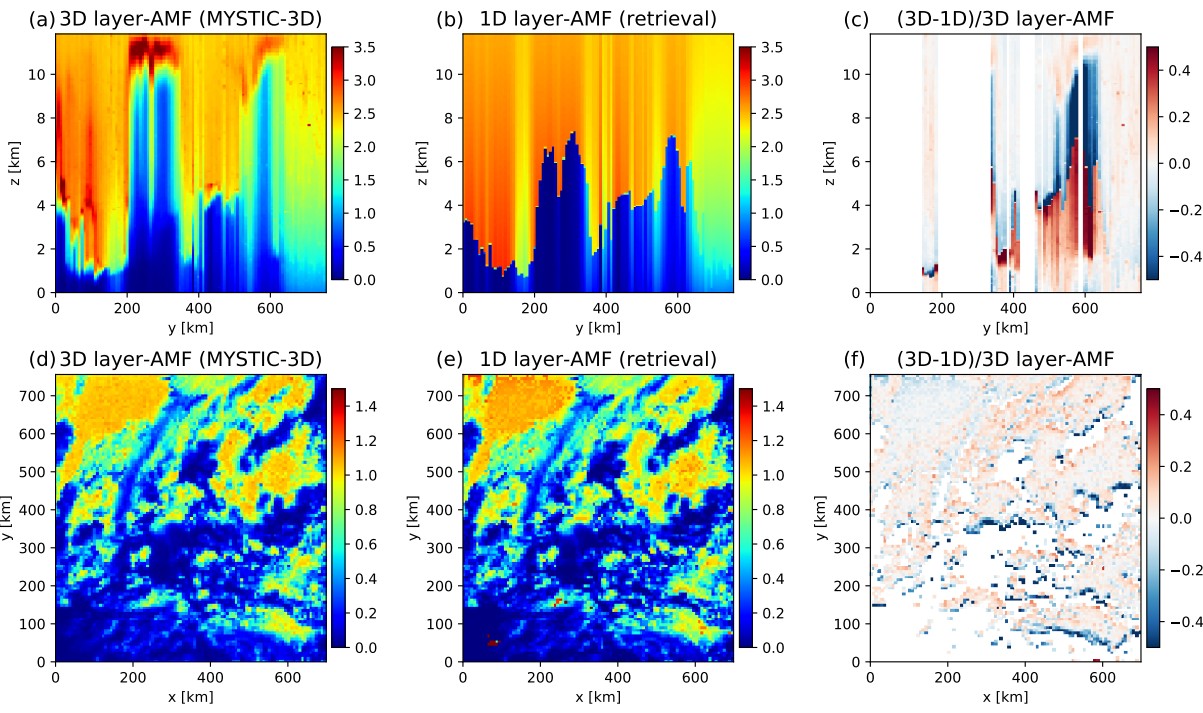

**Figure 13.** Panel (a) shows the vertical cross section of layer-AMFs calculated using 3D radiative transfer with the full cloud information as input (cross section at x=256.8 km). Panel (b) shows the corresponding layer-AMF calculated within the retrieval algorithm using 1D radiative transfer where the cloud is approximated as a Lambertian surface at the effective cloud height. Panel (c) shows the relative difference between 3D and 1D layer-AMF. Panels (d), (e) and (f) show the same layer-AMFs as in the panels (a)–(c) but for a horizontal cross section at 0.5 km altitude.

and liquid clouds mostly below. The effective cloud height gives the altitude at which a Lambertian reflector would produce approximately the same radiance as the clouds.

Panel (d) shows the weighted radiometric cloud fraction (Kylling et al., 2021, Eq.4) derived from the reflectance data. This should not be understood as a pure geometrical cloud fraction. The $NO_2$ retrieval algorithm is only applied to pixels with weighted radiometric cloud fraction less than 0.3. Since the computational time of Monte Carlo radiative transfer simulations depends strongly on the optical thickness of the medium, pixels with high cloud fraction require much more computational time than clear-sky pixels or pixels with low cloud fraction. In order to save computational time, the cloud mask shown in panel (e) was introduced. For all other cases contained in the synthetic dataset, radiances and layer-AMFs were only simulated for pixels with weighted radiometric cloud fraction less than 0.3. Please note that nevertheless the complete LES cloud field is always included as input into the radiative transfer model, consequently pixels near masked thick cloudy pixels can still be affected by those thick clouds. Panel (f) shows the $NO_2$ retrieval error, which can easily be calculated since the "true" input $NO_2$ profile is known and constant over the whole domain.

The synthetic dataset includes 3D layer-AMFs simulated with MYSTIC based on the full LES cloud field, i.e. photons that are scattered from far away clouds into the pixel of interest contribute to the results. Panel (a) in Fig. 13 shows 3D layer-AMFs for the same vertical cross section as in Fig. 12 (yellow line in reflectance image). The largest values are found in the thick cirrus cloud around y=220 km. Panel (b) shows layer-AMFs calculated within the retrieval algorithm which uses a 1D radiative transfer code and a simple approximation of a cloud layer with top altitude derived from the $O_2$-$O_2$ band. In the retrieval algorithm the cloud itself is modelled as a Lambertian surface at the effective cloud height, and clear-sky and cloudy simulations are combined using the retrieved weighted radiometric cloud fraction to obtain approximated layer-AMFs. Panel (c) shows the difference between the 3D layer-AMFs and the approximated 1D layer-AMFs for all pixels with cloud fraction smaller than 0.3 for which the $NO_2$ retrieval is performed. Differences are large (more than 50%) at altitudes within or above the clouds. Since the 1D layer-AMF is used to derive the $NO_2$ concentration in the retrieval the relative difference between "true" layer-AMF and "retrieved" layer-AMF corresponds to the relative retrieval error of the layers of the $NO_2$ concentration profile. If most of the $NO_2$ is situated near the surface, the retrieval error of the VCD will be dominated by layer-AMF differences near the surface. Panels (d), (e), and (f) in Fig. 13 show the layer-AMFs (3D and 1D) and the relative difference between 3D and 1D for a horizontal cross section at 0.5 km altitude. Here, the largest negative differences are found in pixels near thick clouds obviously affected by cloud shadows.

### 4.2.5 Example for $O_2$A-band

For the FRESCO cloud algorithm, averaged reflectances over three bands in the $O_2$A absorption band region are used. As shown in Sec. 4.2.1, these band averages may be calculated sufficiently accurate with the REPTRAN absorption parameterization in fine spectral resolution. In the following, an example simulation for a sensor on a geostationary orbit is presented. In this simulation the solar zenith angle was $40°$, and the solar azimuth angle was $315°$. The viewing zenith angle was $58.3°$ and the viewing azimuth angle $196.3°$. The surface albeo was set to 0.2.

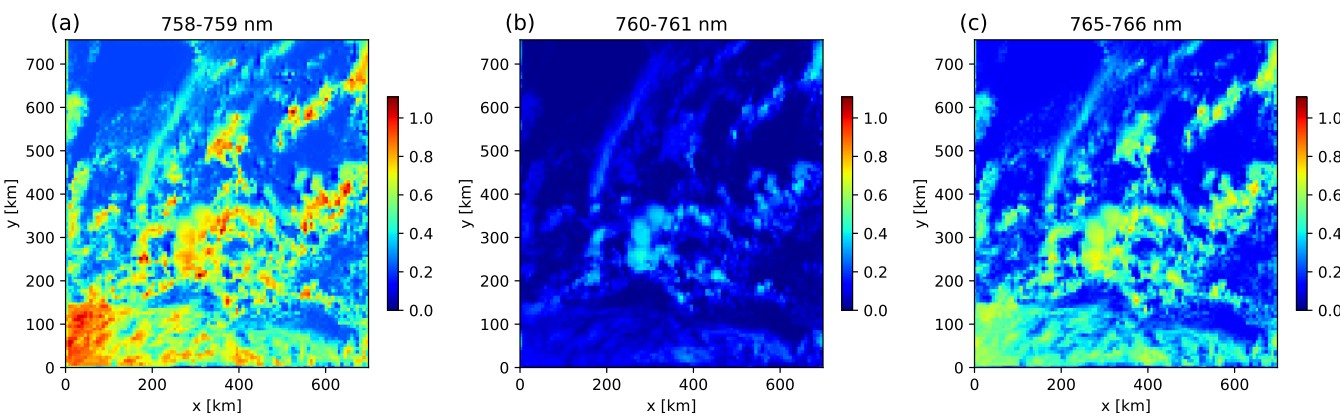

**Figure 14.** Band averaged reflectance simulations in $O_2$A-band region: (a) 758–759 nm, (b) 760–761 nm, (c) 765–766 nm.

|  | Geostationary Orbit | Low Earth Orbit |
|---|---|---|
| solar zenith angles [°] | 20,40,60 | 20,40,60 |
| solar azimuth angles [°] | -90, 45,0,45,90 | 13, 353 |
| sensor viewing zenith angle [°] | 58.3 | 0,20,60 |
| sensor viewing azimuth angle [°] | 196.3 | 109.5, 281.7 |
| surface albedo | 0,0.05,0.2, (0.5 for $O_2$A-band) | |

**Table 2.** Representative sun positions, sensor viewing directions and surface albedos included in synthetic dataset.

Fig. 14 shows the band averaged reflectance images. Panel (a) depicts the average reflectance over 758–759 nm, in the continuum region outside the $O_2$A-band. Panel (b) shows the average over 760–761 nm, in the center of the $O_2$A-band. Panel (c) shows averaged reflectances over 765–766 nm, this range is in the wing of the $O_2$A-band. As expected, highest reflectance values are obtained in the non-absorbing region (a) and lowest values in the center of the absorption band (b).

In panel (b) the brightest pixels are found for the high convective clouds (around $(x,y)$=(300 km, 300 km)), whereas in panel (a) the brightest pixels are found for the thick stratocumulus cloud in the lower left part of the image. This demonstrates the well-known and frequently used feature of the $O_2$A-band, i.e., that it provides information about cloud top height.

### 4.3 Synthetic dataset

In the following the settings and parameters covered by in the synthetic dataset are summarized. For the visible band, the
synthetic dataset includes reflectance spectra from 400–500 nm at a spectral resolution of 0.2 nm, and in addition layer-AMF profiles at 460 nm. Three surface albedos (0, 0.05 and 0.2) were included as suggested in Section 4.2.3. For the $O_2$A-band the synthetic dataset includes averaged reflectance values for three bands: 758–759 nm, 760–761 nm and 765–766 nm. Further, layer-AMFs at 758 nm are provided. Since in the spectral region of the $O_2$A-band, the surface albedo can be significantly higher than in the visible range, in addtion to 0, 0.05 and 0.2, an albedo of 0.5 was added. Representative sun-observer geometries
were chosen based on the considerations in Section 4.2.2. All settings are summarized in Table 2. Overall, the dataset for the visible band includes 45 parameter combinations for geostationary orbit and 108 combinations for Low Earth Orbit. For the $O_2$A-band with an additional surface albedo, 60 combinations for geostationary orbit and 144 combinations for Low Earth Orbit are included. A first detailed analysis of the $NO_2$ retrieval error due to cloud scattering using the synthetic dataset is presented in Kylling et al. (2021).

## 5   Conclusions

A comprehensive synthetic dataset has been generated by simulating satellite observations in two spectral ranges typically used for trace gas remote sensing, the visible range from 400–500 nm and the $O_2$A-band region from 755–775 nm, which is often used for cloud correction. The simulations were performed with the Monte Carlo radiative transfer model MYSTIC, using the ALIS method, which enables efficient simulations at very high spectral resolution. Besides synthetic reflectance spectra,

the dataset includes layer-AMFs, calculated in a 3D atmosphere and thereby including effects of clouds (in-scattering and shadowing) from neighboring pixels.

The dataset includes two cloud-setups: First, simple box-clouds with varying cloud optical thicknesses, cloud heights and geometrical thicknesses were included in a homogeneous atmospheric background and simulations were performed for various sun positions, viewing directions, and surface albedos. Corresponding simulations for clear-sky and 1D clouds are also included. The impact of 3D cloud scattering is quantified by comparing 3D against 1D simulations and the sensitivity on the various cloud parameters, sun-observation geometries, and surface albedos are investigated. In the clear region, the largest impacts are found in the cloud shadow region (typically more than 40% difference between 1D and 3D reflectances). The in-scattering near the cloud edge typically leads to a reflectance increase of about 15%. Further, a comparison between layer-AMFs calculated with horizontal cloud scattering and without is presented. Since trace gas retrieval algorithms usually apply layer-AMFs calculated by a 1D radiative transfer model, this comparison can immediately be mapped to the retrieval error of VCD for a given altitude profile of $NO_2$. This method has been applied to investigate the impact of the profile shape on the retrieval error, largest impacts were found when the $NO_2$ is located at low altitudes.

The second part includes realistic clouds from an ICON Large Eddy Simulation over Europe. The simulated scene includes all cloud types that are typically found in central Europe, such as convective cloud cells, shallow cumulus, stratocumulus and cirrus clouds. An artificial satellite image with a high spatial resolution of $1.2 \times 1.2 \, km^2$ was simulated. Various cloud metrics derived from the artificial image were compared to those derived from real VIIRS satellite images to ensure that the LES clouds are realistic. As for the box-cloud cases, the dataset for the LES clouds includes reflectance spectra in the ranges from 400–500 nm and in the $O_2A$-band region. Representative sun-observer geometries for Low-Earth-Orbits and for geostationary orbits are considered, further also representative surface albedo. The $NO_2$ profile was kept constant over the whole domain.

The synthetic dataset including the box-clouds has been used for a sensitivity study to investigate the impact of the various cloud parameters on the $NO_2$ VCD retrieval error. The most significant bias with several tens of percent was found for cloud shadow regions in polluted areas. The bias depends strongly on various parameters: the solar zenith angle, the cloud optical thickness, the cloud height and the cloud shadow fraction. Based on this sensitivity study, several approaches to correct the retrieval in cloudy conditions have been developed and explored. In order to validate the performance of the retrieval algorithms with cloud correction, these have been applied to the synthetic data for the realistic LES clouds. The result was that using air mass factors calculated with a fitted surface albedo or air mass factor corrected by the $O_2$-$O_2$ slant column density can partly mitigate cloud shadow effects. The details of this investigations are published in the second paper of this series (Yu et al., 2021).

The dataset based on LES clouds has been used to quantify the $NO_2$ VCD retrieval error. An operational retrieval algorithm was applied on the synthetic observations and the retrieval results were compared against the true $NO_2$ VCD which is the knwon model input. The exemplary results show underestimations of the retrieved $NO_2$ VCD in cloud shadow regions of more than 20% and overestimations of about the same order of magnitude in in-scattering regions for the specific sun-observer geometry. In the third paper of the series (Kylling et al., 2021), an analysis of the complete dataset is presented. Cloud shadow fraction, cloud top height, cloud optical thickness, $NO_2$ profile, solar zenith and viewing angle have been identified as the most

important parameters determining the impact of cloud scattering on the $NO_2$ VCD retrieval. For low-earth and geostationary orbit geometries, 89 and 93%, respectively, of the retrieved $NO_2$ VCD were within 10% of the actual VCD for solar zenith angles less than $60°$. For a solar zenith angle of $60°$ the numbers decrease to 53 and 61%. It was also found that for solar zenith angles less than $10°$, the $NO_2$ VCD retrieval error is generally smaller than 10%. For larger solar zenith angles the retrieval error increases to values of the order of tens of percent.

The synthetic dataset may further be used to validate the various different operational trace gas retrieval approaches for Sentinel-S5P. The algorithms can be applied on the synthetic data. Comparing the retrieved $NO_2$ VCDs to the true value used as input to the radiative transfer simulations yields the retrieval accuracy of each algorithm.

*Code availability.* The libRadtran software used for the radiative transfer simulations is available from www.libradtran.org.

*Data availability.* The complete synthetic dataset (in netcdf-format) and corresponding plots of reflectance images, spectra and layer-AMFs are available at https://doi.org/10.5281/zenodo.5567616 (Emde, 2021).

## Appendix A:  Comparison of synthetic and VIIRS/TROPOMI cloud data

As mentioned above the LES results have been validated against ground and satelllite-based observational data by Heinze et al. (2017). In addition we have calculated various cloud metrics for the MYSTIC/LES synthetic simulations and the VI-IRS/TROPOMI data to ensure that the synthetic data are within the variability found in the real data. For the comparison the study region was divided into nine sub-regions has shown in Fig. A1.

For each sub-region the cloud geometric and radiance fractions, number of clouds, and the H-metric were calculated using the VIIRS pixels within one TROPOMI pixel. In addition the distance to the nearest cloud from a VIIRS pixel was estimated. An example of these for measured data are shown in Fig. A2 for one sub-region and one date. The RGB, panel (a) in Fig. A2, shows scattered clouds and large cloud less areas. The clouds are identified with the VIIRS cloud mask, panel (b), which has the TROPOMI pixel grid overlaid the VIIRS cloud mask. The cloud radiance fraction within each TROPOMI pixel is given in panel (c). The cloud geometric and radiance fractions probability distributions are given in panel (d). For this date and sub-region most S5P pixels are nearly cloud free, $CF_r$, $CF_g < 0.1$. For other sub-regions (and dates) this differs considerably (not shown). The H-metric for each TROPOMI pixel is shown in panel (e) and the corresponding histogram for various cloud radiance fraction bins in panel (f). The H-metric is small for cloud less pixels, $CF_r < 0.1$, and cloudy pixels ($CF_r > 0.8$). This is to be expected as radiance variability should be small if either the reflectivity of the surface is homogeneous for the wavelength used to calculate the H-metric, or the cloud is fully covering the TROPOMI pixel. For $0.1 < CF_r < 0.8$ the H-metric is typically significantly larger. Panel (g) plot shows the distribution of the number of clouds within the TROPOMI pixels. Most TROPOMI pixels include one sub-pixel sized cloud, but there are also those that have up to 5 clouds. The

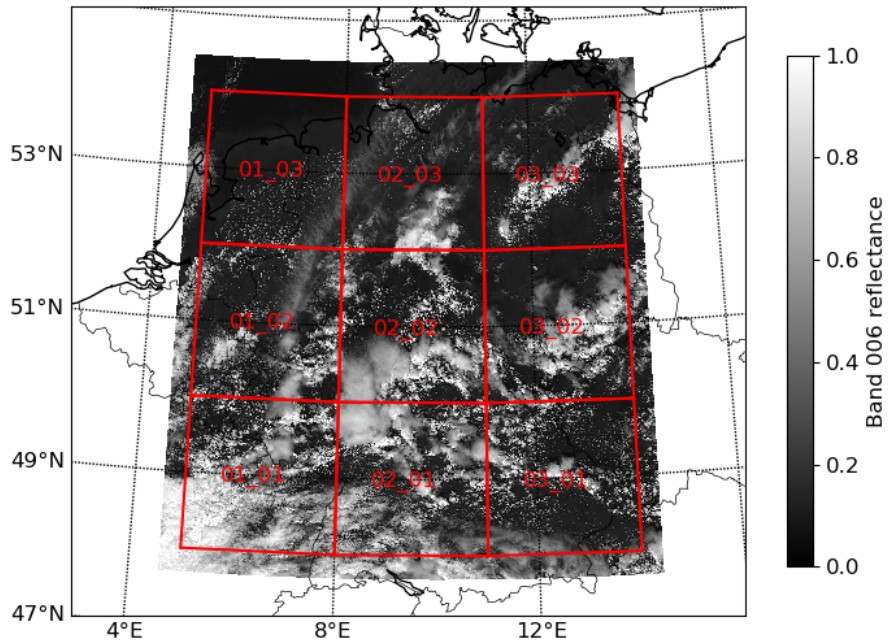

**Figure A1.** MYSTIC simulation of the reflectance for Sentinel3-SLSTR band 1 centred at 555 nm using LES cloud as input, see Section 4 for details. The red rectangles with labels identifies the sub-regions the study area is divided into.

distribution of distances from each VIIRS pixel to the next VIIRS pixel with a cloud is shown in panel (h) in Fig. A2. Finally, the $NO_2$ vertical column density retrieved from the TROPOMI measurements is shown in panel (i).

The mean and standard deviations of the cloud geometric fraction, number of clouds and average cloud size for each sub-region for the study period are given in Table A1. The mean cloud geometric fraction is between 0.48 and 0.54 with standard deviations around 0.31. The mean number of clouds varies from 654 to 979 with standard deviations between 489 and 781. The mean cloud size varies a lot with large standard deviations. The median values are also included in the table and are typically smaller than the means, especially for the cloud size.

To calculate similar synthetic metrics we used MYSTIC to simulate satellite images using the reponse function of SLSTR band 1 centred at 0.555 $\mu$m. It is noted that VIIRS band M4 is centred at the same wavelength. The synthetic images were divided into the 3x3 sub-regions as shown in Fig. A1. An example of synthetic cloud metrics for the 02_02 sub-region is given in Fig. A3. While there are similarities between the synthetic based results and those presented in Fig. A2 for VIIRS/TROPOMI, there are also differences. The H-metric has a slightly larger span in the synthetic data including higher H-metric values for CRF between 0.2-0.4. However, the synthetic results presented in Fig. A3 appears to be realistic and representative for the real conditions as observed by VIIRS and TROPOMI.

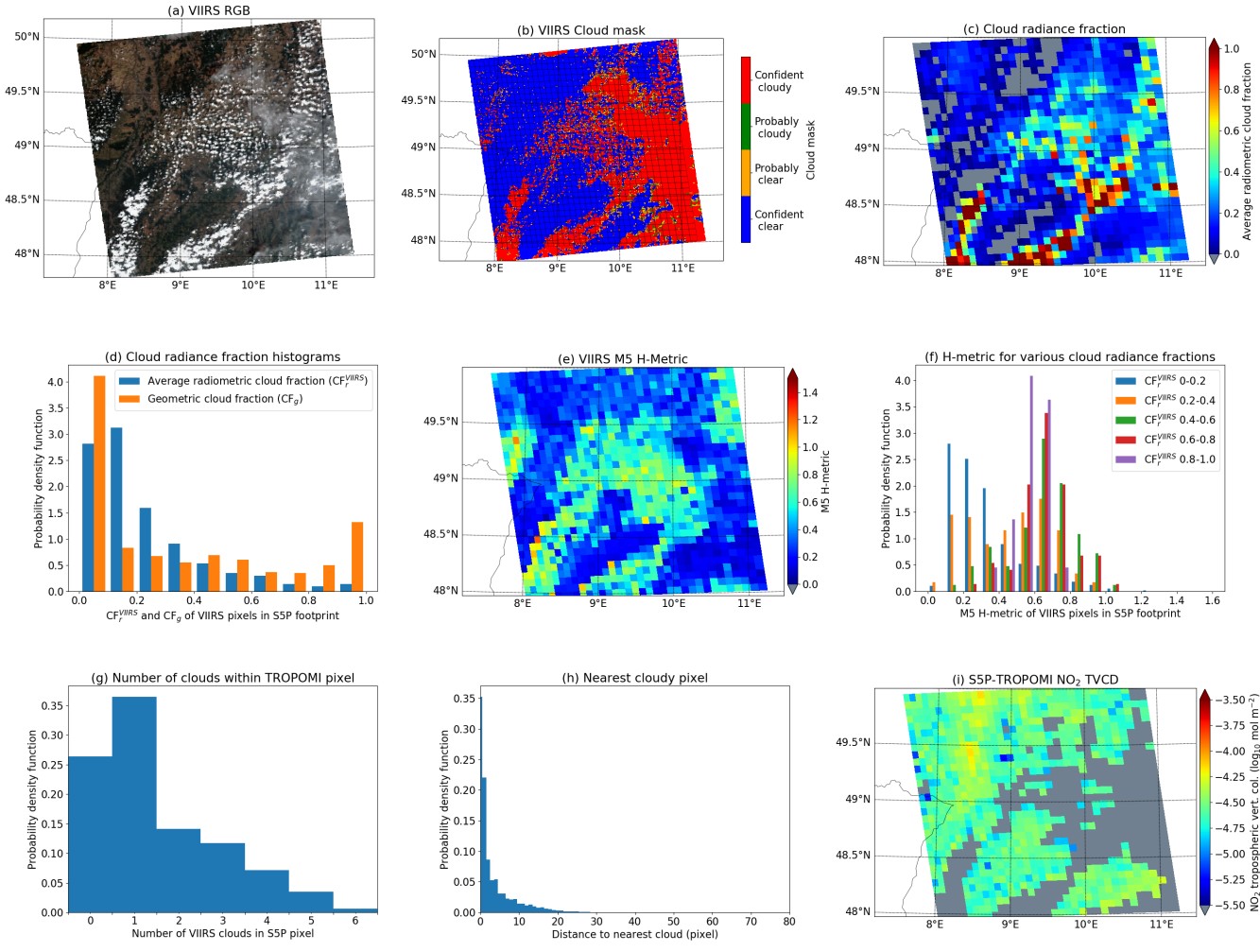

**Figure A2.** Examples of VIIRS data and cloud metrics for TROPOMI pixels for sub-region 02_01 and 2018/8/5. (a) VIIRS RGB. (b) VIIRS cloud mask with TROPOMI grid overlaid. (c) Cloud fraction for TROPOMI pixels. (d) Distribution of cloud fraction. (e) H-metric for TROPOMI pixels. (f) Distribution of H-metric for various cloud fractions. (g) Distribution of number of clouds within TROPOMI pixels. (h) Distribution of distance to nearest cloudy pixel. (i) NO$_2$ column as retrieved from TROPOMI data. Gray pixels indicate data with low quality flag.

In Table A2 statistics for the cloud metrics are presented for the synthetic data. Comparing the synthetic statistics in Table A2 with the VIIRS/TROPOMI statistics presented in Table A1, it is noted that several sub-regions are within the median of the VIIRS/TROPOMI cloud metric statistics for the cloud fraction and cloud size, and median number of clouds. Thus, both quantitatively and qualitatively the synthetic data capture the observed variability in the cloud metrics. This, together with the

**Table A1.** Averaged cloud metrics calculated from the VIIRS data for the study period. Cloud size is in units of VIIRS pixels.

| Region | $CF_r$ mean±std | $CF_r$ median | $CF_g$ mean±std | $CF_g$ median | # Clouds ±std | # Clouds median | Mean cloud size ±std | Cloud size median |
|---|---|---|---|---|---|---|---|---|
| 01_01 | 0.33±0.25 | 0.25 | 0.48±0.32 | 0.46 | 979±782 | 862 | 4546±21632 | 38 |
| 01_02 | 0.32±0.24 | 0.26 | 0.51±0.32 | 0.51 | 837±708 | 615 | 664±2865 | 63 |
| 01_03 | 0.33±0.25 | 0.26 | 0.54±0.31 | 0.61 | 654±489 | 518 | 3489±19150 | 80 |
| 02_01 | 0.36±0.27 | 0.27 | 0.50±0.31 | 0.43 | 970±779 | 825 | 3729±18876 | 41 |
| 02_02 | 0.34±0.27 | 0.27 | 0.51±0.33 | 0.46 | 823±732 | 579 | 2328±12236 | 37 |
| 02_03 | 0.34±0.26 | 0.26 | 0.51±0.30 | 0.49 | 809±581 | 718 | 2784±15876 | 53 |
| 03_01 | 0.38±0.28 | 0.28 | 0.54±0.30 | 0.52 | 815±678 | 678 | 4124±16455 | 57 |
| 03_02 | 0.38±0.25 | 0.25 | 0.54±0.30 | 0.54 | 795±634 | 700 | 1687±8505 | 61 |
| 03_03 | 0.34±0.24 | 0.24 | 0.50±0.31 | 0.47 | 839±655 | 740 | 2628±14961 | 46 |

**Table A2.** Cloud metrics for each sub-regions calculated from synthetic data. Note that no standard deviation is provided as only one synthetic based image is used to calculate the cloud metrics. Cloud size is in units of VIIRS pixels.

| Region | $CF_r$ mean | $CF_r$ median | $CF_g$ mean | $CF_g$ median | # Clouds | # Clouds median | Mean cloud size | Cloud size median |
|---|---|---|---|---|---|---|---|---|
| 01_01 | 0.65 | 0.77 | 0.75 | 0.75 | 331 | 331 | 185 | 185 |
| 01_02 | 0.55 | 0.55 | 0.66 | 0.66 | 516 | 516 | 104 | 104 |
| 01_03 | 0.25 | 0.14 | 0.35 | 0.35 | 965 | 965 | 30 | 30 |
| 02_01 | 0.27 | 0.15 | 0.41 | 0.41 | 714 | 714 | 47 | 47 |
| 02_02 | 0.42 | 0.30 | 0.50 | 0.50 | 543 | 543 | 75 | 75 |
| 02_03 | 0.33 | 0.11 | 0.42 | 0.42 | 402 | 402 | 86 | 86 |
| 03_01 | 0.04 | 0.00 | 0.10 | 0.10 | 743 | 743 | 10 | 10 |
| 03_02 | 0.14 | 0.01 | 0.28 | 0.28 | 414 | 414 | 54 | 54 |
| 03_03 | 0.16 | 0.00 | 0.21 | 0.21 | 568 | 568 | 30 | 30 |

independent validation by Heinze et al. (2017), gives confidence to the assumption that the $NO_2$ -bias estimates based on the synthetic results are representative for the real atmosphere.

*Author contributions.* CE is the main contributor to the study. She performed the radiative transfer simulations to generate the synthetic datasets, analyzed the impact of 3D cloud scattering, and she led the writing of this paper.

HY applied the $NO_2$ retrieval algorithm on the synthetic data and provided the example results.

AK calculated cloud metrix from the synthetic data and compared them to those derived from satellite observation. Further he extracted representative sun-observer geometries for LEO and GEO orbits, as well as representative surface albedos.

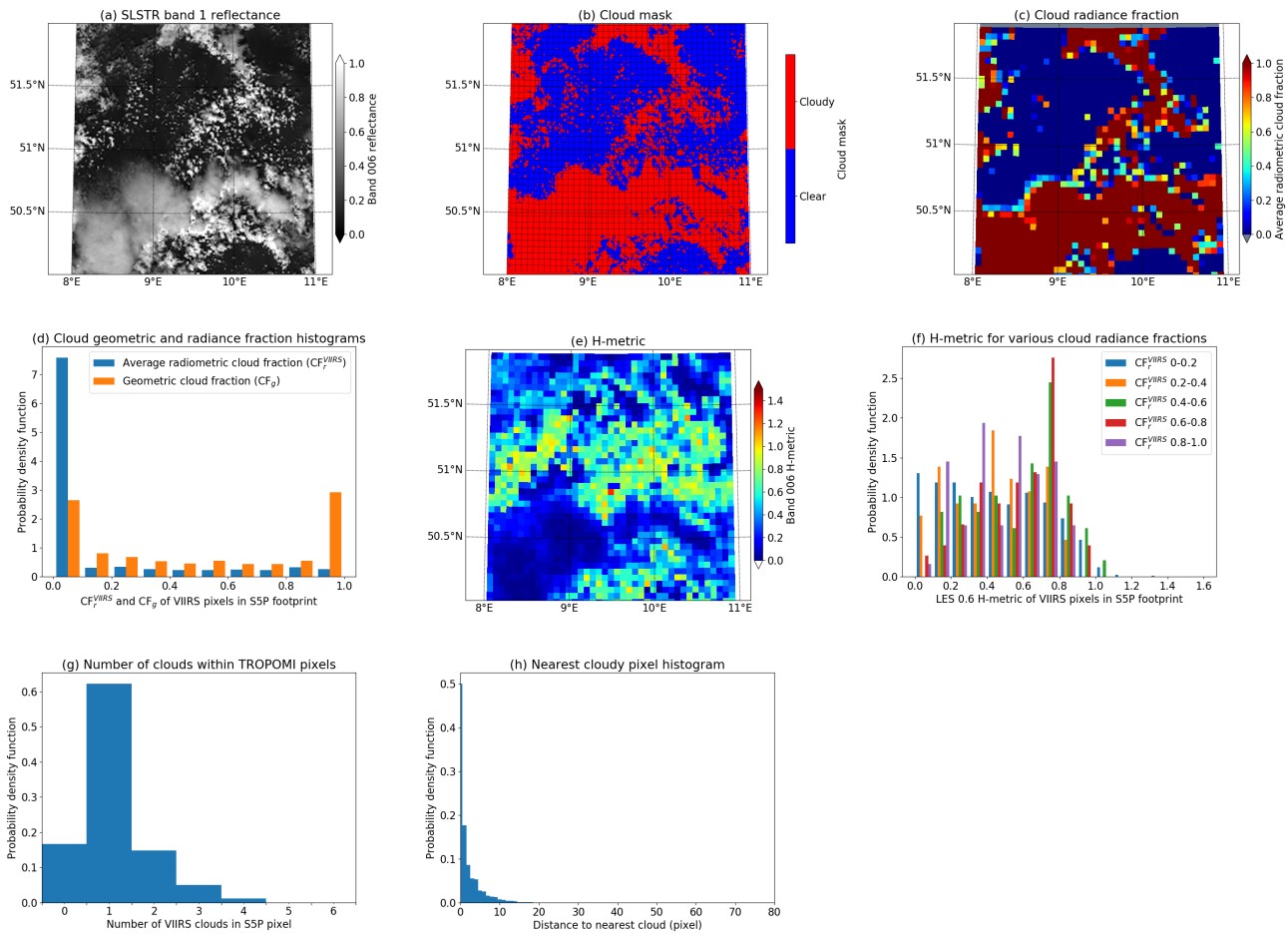

**Figure A3.** Examples of LES data and cloud metrics for TROPOMI pixels for sub-region 02_02. (a) Simulated SLSTR band 1 reflectance. (b) LES cloud mask with TROPOMI grid overlaid. (c) Cloud radiance fraction for TROPOMI pixels. (d) Distribution of cloud geometric and radiance fractions. (e) H-metric for TROPOMI pixels. (f) Distribution of H-metric for various cloud radiance fractions. (g) Distribution of number of clouds within TROPOMI pixels. (h) Distribution of distance to nearest cloudy pixel.

MvR, KS, BV and BM contributed to conceptualization and methodology.

*Competing interests.* The authors declare that no competing interests are present.

*Acknowledgements.* This work was funded by ESA (3DCATS project 4000124890/18/NL/FF/gp). We thank Leonhard Scheck for providing the ICON cloud field data in MYSTIC input format.

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
