# Peer review of "Impact of 3D Cloud Structures on the Atmospheric Trace Gas Products from UV-VIS Sounders – Part I: Synthetic dataset for validation of trace gas retrieval algorithms"

_Atmospheric Measurement Techniques, 2021_

## Referee Comment (RC1)

Review of "Impact of 3D Cloud Structures on the Atmospheric Trace Gas Products from UV-VIS Sounders – Part I: Synthetic dataset for validation of trace gas retrieval algorithms" by Emde et al.

This paper is one of a set of three interconnected papers that discusses a) a publicly available synthetic dataset of 3D radiances, b) the sensitivity of vertical column density NO2 retrieval errors near box-clouds and observations, and c) 3D cloud biases and metrics. The reviewed paper is part a) of the full set of papers.

Since the paper is the first of a three-set collection, main findings are reserved for the other two papers. This produces the awkward situation that the main physical results, which one can derive from an analysis of the synthetic dataset, are not discussed in the reviewed paper. The reviewed paper is overly restrained. The reviewed paper is rather short of main findings, mainly stating that a synthetic data set is available, and therefore limited in informative results.

There are places in the text in which a terse one sentence paragraph is stated. Additional sentences can and should be added to the text in these portions of the text.

The paper should be revised. A suggested addition to the revised paper would be to include a figure or two that demonstrates the in-scattering and shadow curves (similar to Figure 2) for clouds of e.g. three heights in the LES cloud field, including panels in which (on the y axis) the reflectance is graphed as a function of distance from cloud edge, and panels in which retrieved NO2 is graphed (on the y axis) as a function of distance from cloud edge. These figures would help the research community better appreciate the quantitative importance of 3D cloud effects upon NO2 retrievals. Over what km range are 3D effects present as a function of cloud height, and what are the % errors in the NO2 column for these situations?

The paper is worthy of publication following revision.

Major comments

I would have liked to have seen in the Conclusions section a discussion of the major physical findings of the paper. The Figures provide instructive insights, yet these insights are barely touched upon in the Conclusions.

Minor comments

Page 3, line 12. Clarify what is meant by "bias" (the bias of what?)

Page 3, line 12. Replace by "In the third paper by Kylling..". The one sentence paragraph is a bit jarring since it is overly short in informative content.

Page 4, line 3. Clarify what is meant by "unbiased radiances"

Page 4, line 6. Replace "agreed perfectly" with a quantitative % value.

Page 4, lines 14-17. I am not convinced of the ability of the authors to "calculate the full spectrum based on photon path distributions sampled at a single wavelength". Atmospheric optical properties (Rayleigh scattering, aerosol optical depths, asymmetry and single scattering albedo) have a wavelength dependence in a real atmosphere. Please support your statements in the context of a real atmosphere with additional sentences. The sentence "The statistical error of

such a simulation is a bias for the complete spectrum" is not comprehensible. Add additional sentences which discuss the ALIS method. Replace with "This method allows one to calculate .."

Page 5, line 1. What is the boundary layer height, and how is NO2 vertically distributed in the troposphere?

Page 5, line 10. Change to "in the x-direction".

Page 5, line 14. The sentence implies just a single box-cloud geometry, while Table 1 lists several box-cloud heights. Rephrase to "For the liquid water cloud the primary cloud geometry has the base height set to 2 km …".

Page 6, line 4. Why were aerosols not included in the calculations?

Page 6, line 11. Clarify what is meant by "variance reduction methods". Add sentences that describe the VROOM methods.

Page 9, line 12. Change to "Note that $D(\lambda)$ is a smooth function"

Page 11, line 8. Change to "layer-AMF as a function of"

Pager 13, line 2. Change to "realistic 3D clouds"

Page 15, line 17. Replace "sufficiently accurate" with a quantitative % accuracy.

Page 15, line 22. Change to "was analyzed, and it was found that SZA.."

Page 16, line 1. Change to "It was found that SZA varies.."

Page 17, line 1. Change to "Note that each simulated pixel includes 36 cloud pixels,.."

Page 17, line 5. Change to "Note that the number of .."

Page 18, Figure 11. What are the units of the NO2 retrieval error?

Page 19, line 16. Change to "Note that the complete LES.."

Page 23, line 7. Explain how the synthetic dataset can be used to "validate the various different trace gas retrieval approaches for Sentinel-S5P." This is an example of a terse one sentence paragraph that would benefit from additional sentences.

1. Does the paper address relevant scientific questions within the scope of AMT? yes

2. Does the paper present novel concepts, ideas, tools, or data? The box-cloud and LES 3D calculations are fairly unique

3. Are substantial conclusions reached? No (more discussion would add to the paper)

4. Are the scientific methods and assumptions valid and clearly outlined? Some additional sentences (e.g. on the ALIS method) should be added to the text

5. Are the results sufficient to support the interpretations and conclusions? yes

6. Is the description of experiments and calculations sufficiently complete and precise to allow their reproduction by fellow scientists (traceability of results)? yes

7. Do the authors give proper credit to related work and clearly indicate their own new/original contribution? yes

8. Does the title clearly reflect the contents of the paper? yes

9. Does the abstract provide a concise and complete summary? yes

10. Is the overall presentation well structured and clear? Some additional sentences should be added to the text to clarify the discussion

11. Is the language fluent and precise? yes

12. Are mathematical formulae, symbols, abbreviations, and units correctly defined and used? yes

13. Should any parts of the paper (text, formulae, figures, tables) be clarified, reduced, combined, or eliminated? There are several places in the text in which clarifications are suggested

14. Are the number and quality of references appropriate? yes

15. Is the amount and quality of supplementary material appropriate? Not applicable

---

## Referee Comment (RC2)

**Review "Impact of 3D Cloud Structures on the Atmospheric Trace Gas Products from UV-VIS Sounders – Part I: Synthetic dataset for validation of trace gas retrieval algorithms" by Emde et al.**

The manuscript by Emde et al. (2021) presents a synthetic dataset of 3D radiative transfer simulations that can be used to investigate the effect of 3D cloud structures on trace gas retrievals. This is a relevant scientific topic, particularly in view of the increased spatial resolution of current and future satellite instruments. The effects of realistic 3D cloud structures have not been yet extensively investigated and quantified, so this study (and its companion papers) are of scientific significance. However, the presentation quality is just sufficient, and although the scientific approach and methods are valid, the results are not discussed in an appropriate way. Therefore, the manuscript is well suited for publication, but major revisions are needed.

**General comments**

The manuscript reads often like a technical report, sometimes difficult to follow. The fact that it presents a synthetic dataset as a part of three publications decreases the possibility of more in depth and detailed discussions. However, scientific discussion needs to be more extensive, otherwise it could just be published in a journal more suitable for datasets.

Two main issues that are of central importance (together with clouds) to trace gas retrievals are aerosols and NO2 profile. The aerosols and its correction on the retrieval algorithms is closely related to the clouds and their effects. Furthermore, the study uses the most realistic cloud field possible with 3D radiative transfer model but then uses a constant NO2 field throughout the domain. These two topics need to be discussed in the text, as currently is just mentioned as if this was of very little relevance.

**Specific comments**

Introduction:
The first two-three paragraphs as compared to the rest of the introduction are poorly written. The last sentence of the first paragraph does not add anything to the readability of the introduction and to the topic of the manuscript.

Page 2, lines 24-26: 'Here' referring to Schwaerzel et al? What do you mean by 3D box-airmass-factors? Calculated with 3D radiative transfer model? Please be more specific. This is the introduction and it is already somewhat confusing the naming of the AMFs. Later on, you also refer to 3D layer-AMFs (e.g. page 19, line 20), so please be consistent.

Page 3, line 33: what do you mean by inhomogeneous surface albedo? Spectral dependence? Reflection anisotropy?

Page 4, line 15: is the ALIS method influenced by the number of photons on the simulation?

Page 4, line 18-19: see my comment on the introduction about layer and box AMFs.

Last sentence on Sect. 2: maybe you can cite the paper of the series where this is actually analyzed. In this manuscript only one specific layer AMF has been analyzed (at 0.5 km).

Sect. 3:
Page , line 1: 'most of the NO2 located within the BL'. How high is the boundary layer for your base case? Have you investigated the effects in a non-polluted atmosphere? In connection to the apriori profile and the horizontal effects; the TROPOMI NO2 bias as reported by validation studies is different for urban and rural areas, so the 3D clouds effect may play a different role in these biases depending on the pollution level. See also your sentence in page 12, line 11. This needs further discussion, 'more or less affected' is not rigorous.

Later on the section it is also mentioned that 'Aerosols are not included'. Aerosols are a relevant topic for NO2 retrievals, so it needs further discussion. Even if it is not included, some motivation for this decision should be discussed, as well as the effect that the inclusion of an explicit aerosol treatment would have in the results.

What is the vertical discretization of the atmosphere? How does affect your simulations?

Sect. 3.1.2: what do you mean that in the y-direction the cloud layer extended to infinity? As I later understand the cloud has a geometrical dimension, right?

If half of the domain extending from 0 to 100km is cloud-free and the other half has a cloud, this does not match the schematic in Fig. 1.

Page 6, line 3: above the cloud as in the vertical dimension?

Page 7, line 7: 'the reflectance is higher than the clear sky reflectance near the cloud edge.'. Reads weird, please rephrase.

Page 6, line 7: 1D cloud layer setup. This is the first time this is mentioned. Do you mean just a cloud acting as a Lambertian reflector? Please explain.

Page 6, line 11: what are variance reduction methods, why are they needed? Please explain.

Reading of figure 1 could benefit from the addition of a grid. On page 9, line 31: do you mean larger than -15%?

Sect. 3.2.3. The explanation on this section could benefit from an Eq. that shows how the AMF is used in the retrieval.

Last sentence in page 12: it would be beneficial to include a discussion with few sentences on the main findings even though they are published in Yu et al.

Page 15 line 29: TROPOMI was not launch until October 2017.

Sect. 4.2.3: I would suggest to substitute the global maps of surface albedo for a zoom over the study for which the cloud simulations are done.

Sect. 4.2.4:

What about higher resolution than 7 km x 7 km? This is good for TROPOMI, but future sensors will definitely provide measurements at higher resolution. In Sect. 3 the simulations are done for higher resolution, and the ICON clouds resolution is 1.2 km x 1.2 km. The increased spatial resolution (as pointed in the manuscript) will enhance the impact of 3D clouds effects, so it would benefit the discussion to perform these simulations at higher resolution. If this is not feasible, then at least this should be discussed.

What is the effect on the results of reducing the number of photons with respect to the 1D/3D case? How will this affect the airmass factor calculation?

The study uses a very realistic cloud field from LES simulations, but then assumes a constant NO2 field over the whole domain, which is very unrealistic. The consequences of this assumption on NO2 needs to be discussed. How would the NO2 retrieval error on Fig. 12 look like if a realistic NO2 field would be used?

**Editing**

Figures and figure captions should be revised. Different sub-figures are specified differently in different figures, so please revise. See [https://www.atmospheric-measurement-techniques.net/submission.html](https://www.atmospheric-measurement-techniques.net/submission.html) for figure guidelines. Using letters a,b,c etc. makes referencing on the text easier. Please mind the reader when creating the figures.
For example, Fig. 10: '(similar to TROPOMI on Sentinel-5P and Sentinel-5)' this is not relevant in a figure caption. Lower panels 'x = 256 km' is not relevant information and

makes the figure busier. Another example, Fig. 3 Top: legend 'clear' is better understood than '-1.5' and '-10.5 km', maybe add 'shadow' and 'clear region'.

Name the O2 A band consistently throughout the manuscript (three names O2-A band, O2A-band, O2A band have been used)

Page 2, line 28: for->from TROPOMI/S5P obs.
Page 2, line 29: synthetic -> synthetic
Page 2, line 30: are-> were not included

Page3, line 3: incorrect grammar; the bias due to 3D clouds on what? And no need to start new paragraph if you talk about the same paper.

Page 18, limes 5-10: please write sentences in present tense. E.g., 'pathlength is decreased' -> 'decreases'

Short paragraphs (1-2 sentences) just expressing technical details should be avoided.

Please include (at least) a reference when mentioning FRESCO cloud algorithm.

Page 22, line 6: what about the shadow effects?
Page 22, line 23: was this not at 1.2 km x 1.2 km?

---

## Author Comment (AC2)

**Answers to referee 2 comments "Impact of 3D Cloud Structures on the Atmospheric Trace Gas Products from UV-VIS Sounders – Part I: Synthetic dataset for validation of trace gas retrieval algorithms"**

We thank the reviewer for constructive comments that helped us to improve our manuscript.

In the following we respond point-by-point to all reviewer comments.

**The manuscript reads often like a technical report, sometimes difficult to follow. The fact that it presents a synthetic dataset as a part of three publications decreases the possibility of more in depth and detailed discussions. However, scientific discussion needs to be more extensive, otherwise it could just be published in a journal more suitable for datasets.**

The main scientific findings are included in part II and part III, this is correcrt. However, we think, that the description of how the dataset is created, which methods are used etc. is worth a publication. To our knowledge this is the first synthetic dataset including high resolution spectra (as used for trace gas remote sensing) and three-dimensional realistic clouds, generated using a three-dimensional radiative transfer model.

**Two main issues that are of central importance (together with clouds) to trace gas retrievals are aerosols and NO2 profile. The aerosols and its correction on the retrieval algorithms is closely related to the clouds and their effects. Furthermore, the study uses the most realistic cloud field possible with 3D radiative transfer model but then uses a constant NO2 field throughout the domain. These two topics need to be discussed in the text, as currently is just mentioned as if this was of very little relevance.**

See answers below.

**Introduction: The first two-three paragraphs as compared to the rest of the introduction are poorly written. The last sentence of the first paragraph does not add anything to the readability of the introduction and to the topic of the manuscript.**

Removed the last sentence from the first paragraph. Slightly changed the other two sentences.

**Page 2, lines 24-26: "Here" referring to Schwaerzel et al? What do you mean by 3D box-airmass-factors? Calculated with 3D radiative transfer model? Please be more specific. This is the introduction and it is already somewhat confusing the naming of the AMFs. Later on, you also refer to 3D layer-AMFs (e.g. page 19, line 20), so please be consistent.**

Removed this statement from the introduction and included more detailed definitions of layer-AMF and box-AMF in the model description section. Here we also shorty explain the DOAS method, how the layer-AMFs are used in the retrieval, and how layer-AMFs can be used to study the impact of cloud scattering on the retrieval.

"In the UV and visible spectral ranges, the standard retrieval algorithm is based on the DOAS

technique Platt (2017): in a first step, the slant column density (SCD) is retrieved by spectral fitting of the observed solar spectra to absorption cross sections of trace gases. The SCD corresponds to the amount of trace gas along the average photon path from the Sun through the atmosphere to the satellite sensor. In order to convert SCD into a vertical column density (VCD), the so-called air-mass factor is required, which is defined as the ratio between SCD and VCD. In clean regions, the retrieval error is dominated by the spectral fitting, while for polluted or cloudy regions, the uncertainty of the AMF becomes the dominant error source. The AMF is calculated using radiative transfer models.

MYSTIC includes the option to simulate 1D layer-AMFs or 3D box-AMFs (Schwaerzel et al., 2020). The concept of layer/box-AMFs assumes that the trace gas concentration is small compared to the concentration of other gases, meaning that interaction of photons with trace gas molecules does not alter the photon path distribution in the atmosphere. Layer-AMFs are calculated from the photon path length distribution in each individual altitude layer of the model atmosphere as described in Deutschmann et al. (2011). MYSTIC allows to calculate layer-AMFs for 1D plane-parallel or spherical atmospheres, and also for 3D model atmospheres. In the latter case the photon pathlengths are integrated horizontally over the full domain. Note that these "3D" layer-AMFs still include the impact of 3D cloud scattering. In DOAS type retrievals the layer-AMFs are used together with the a priori $NO_2$ altitude profile to compute the total AMF:

$$\mathrm{AMF} = \frac{\sum_l \mathrm{AMF}_l \cdot x_l}{\sum_l x_l} \tag{1}$$

Here $l$ is the layer index, $\mathrm{AMF}_l$ the layer-AMF and $x_l$ the partial column density for layer $l$. This AMF is then used to convert from slant column density (SCD) to vertical column density (VCD):

$$\mathrm{VCD} = \mathrm{SCD}/\mathrm{AMF} \tag{2}$$

Note that in the literature, layer-AMFs are commonly called box-AMF (e.g. Deutschmann et al. (2011)), which is a confusing terminology, because they do not refer to model grid boxes. MYSTIC also enables the calculation of real "box"-AMFs which are derived from the 3D photon pathlength distribution, i.e. from the photon pathlengths in each 3D model grid cell. Box-AMFs are useful if one knows a 3D a priori $NO_2$ concentration distribution which can be used in the retrieval to convert from SCD to VCD (Schwaerzel et al., 2020). All currently available operational retrieval algorithms apply 1D a priori altitude concentration profiles, therefore they can not use box-AMFs.

Using MYSTIC, we may study how the layer-AMFs are modified by scattering from clouds in the neighborhood. Comparing the layer-AMFs of a clear sky atmosphere with the layer-AMFs influenced by clouds, we may estimate the retrieval error of, e.g., $NO_2$ vertical column densities (VCDs). Working with simulated layer-AMFs allows us also to study the impact of the vertical $NO_2$ concentration profile on the retrieval error. Since the influence of trace gases on the photon pathlength distribution and thus on layer-AMF is negligible, we may use the layer-AMFs of one radiative transfer simulation to estimate the error for various assumed $NO_2$ concentration profiles. Such an analysis is presented in part II of this publication series (Yu et al., 2021). For this reason it is not necessary to include simulations for different $NO_2$ profiles in the synthetic dataset."

**Page 3, line 33: what do you mean by inhomogeneous surface albedo? Spectral dependence? Reflection anisotropy?**

We actually meant a 2D surface albedo map. But spectral dependence and reflection anisotropy can also be handled. Included a more precise description in the text.

**Page 4, line 15: is the ALIS method influenced by the number of photons on the simulation?**

Yes, the bias decreases with the number of photons. This statement has been added to the text.

**Page 4, line 18-19: see my comment on the introduction about layer and box AMFs.**

See answer above.

**Last sentence on Sect. 2: maybe you can cite the paper of the series where this is actually analyzed. In this manuscript only one specific layer AMF has been analyzed (at 0.5 km).**

Yes, a more detailed analysis is presented in part II of the series (Yu et al., 2021). The reference has been added to the text.

**Sect. 3: Page , line 1: 'most of the NO2 located within the BL'. How high is the boundary layer for your base case? Have you investigated the effects in a non-polluted atmosphere?**

We included a figure showing typical NO2 profiles, including the polluted one which was used for the base case. We also investigated effects for a non-polluted atmosphere (actually here the effects are much smaller). This can be done using the layer-AMFs, which do not depend on the NO2 profile, as long as NO2 molecules do not change the photon pathlength distribution. Generally this is a good assumption for trace gases.

We included a justification for including only one NO2-profile in the synthetic dataset: "As mentioned before we may use layer-AMFs to investigate the impact of cloud scattering on the trace gas concentration retrieval. Layer-AMFs are independent of the trace gas profiles, for this reason we define only one $NO_2$-profile, but still we can investigate retrieval errors also for different profiles including non-polluted cases (see also Yu et al. (2021))."

**In connection to the apriori profile and the horizontal effects; the TROPOMI NO2 bias as reported by validation studies is different for urban and rural areas, so the 3D clouds effect may play a different role in these biases depending on the pollution level. See also your sentence in page 12, line 11. This needs further discussion, 'more or less affected' is not rigorous.**

Yes, of course the retrieval error depends a lot on the NO2 profile. We investigated this in detail using the synthetic data. At the end of section 3 we include a brief summary of the sensitivity study based on the synthetic data for the box cloud case which is presented in detail in Yu et al. (part II).

"A detailed sensitivity study of the $NO_2$ retrieval error based on the box-cloud synthetic dataset is presented in Yu et al. (2021). Largest retrieval biases were found in the cloud shadow region, typically the errors are in the range of 10–100% for the polluted scenario. The bias increases with

solar zenith angle, decreases with surface albedo and it increases with cloud optical thickness. The dependency on cloud geometrical thickness and cloud bottom height is less pronounced. Yu et al. (2021) also show that the cloud effects are much stronger for polluted cases compared to non-polluted cases, the maximum retrieval bias for the polluted profile is 95% for the base case settings and for the clean profile it is reduced to 6%. Various different $NO_2$ profile shapes have been investigated in addition, clearly demonstrating that the retrieval bias depends on the altitude where most of the $NO_2$ is located. The synthetic data was also applied to investigate the dependancy of the retrieval bias on the spatial resolution of the instrument. The synthetic data is created for a sensor footprint of $1 \times 1\,km^2$. By averaging, spatial resolutions between 3-15 km could be investigated. As expected, the retrieval bias decreases with increasing spatial resolution due to spatial averaging. The cloud shadow effect strongly depends on the cloud shadow fraction in a pixel."

**Later on the section it is also mentioned that "Aerosols are not included". Aerosols are a relevant topic for NO2 retrievals, so it needs further discussion. Even if it is not included, some motivation for this decision should be discussed, as well as the effect that the inclusion of an explicit aerosol treatment would have in the results.**

Included the following discussion:

"Aerosols were not included, although aerosol scattering also has a significant impact on the $NO_2$ retrieval. However, in this study, we aim to quantify the impact on cloud scattering on the retrieval. When both, aerosols and clouds are included, it becomes difficult to disentangle the impacts of cloud and aerosol scattering. Therefore, we decided to include only clouds."

**What is the vertical discretization of the atmosphere? How does affect your simulations?**

Included the following sentence for clarification:

"We have chosen a fine vertical resoltution of the model atmosphere in the lower part of the atmosphere, between 0 km and 12 km altitude the layer thickness is about 150 m. The vertical resolution from 12–25 km is 1 km, from 25–50 km 2.5 km and from 50–100 km 5 km. We have chosen the fine vertical resolution in the lower part of the atmosphere in order to resolve the vertical dependency of layer-AMF in the region of interest."

**Sect. 3.1.2: what do you mean that in the y-direction the cloud layer extended to infinity? As I later understand the cloud has a geometrical dimension, right? If half of the domain extending from 0 to 100km is cloud-free and the other half has a cloud, this does not match the schematic in Fig. 1.**

MYSTIC uses periodic boundary conditions, this information has been added to the text. It means that next to the cloud, there is again a clear region, so the schematic was in principle correct. However, in order to clarify the setup, we updated the schematic so that half of the domain is clear and half cloudy, and we show the in-scattering and the shadowing geometry separately, using the same cloud definition but changing the sun direction exactly as done in the simulations. The y-direction really extends to infinity as stated in the text.

**Page 6, line 3: above the cloud as in the vertical dimension?**

Changed this formulation to "... starting at a distance of 15 km away from the cloud egde in the

clear region and ending at a distance of 10 km in the cloudy region."

**Page 7, line 7: "the reflectance is higher than the clear sky reflectance near the cloud edge." Reads weird, please rephrase.**

Rephrased to: "In the in-scattering region (left panels), the cloudy reflectance is larger than the clear sky reflectance."

**Page 6, line 7: 1D cloud layer setup. This is the first time this is mentioned. Do you mean just a cloud acting as a Lambertian reflector? Please explain.**

No, we use exactly the same cloud definition (same optical properties, liquid water content etc.) but extend the cloud layer over the full domain. Clarified this in the text.

"For all combinations of parameters we also calculated radiance spectra for a corresponding 1D cloud layer setup, where the cloud is extended horizontally over the full model domain. The cloud optical and microphysical properties are exactly the same as for the 3D cloud simulations."

**Page 6, line 11: what are variance reduction methods, why are they needed? Please explain.**

Included a short explanation: "The variance reduction methods VROOM (Buras and Mayer, 2011), which reduce the statistical noise in Monte Carlo radiative transfer simulations including cloud scattering," Explaining the methods in detail is out of the scope of this paper, the reader needs to refer to the given reference.

**Reading of figure 1 could benefit from the addition of a grid. On page 9, line 31: do you mean larger than -15%?**

We added the x and y axes to figure 1 for clarification. Yes, we mean -15% and corrected this.

**Sect. 3.2.3. The explanation on this section could benefit from an Eq. that shows how the AMF is used in the retrieval.**

We included in the model description part along with the definition of the layer-AMFs the basic equations used in the retrieval to convert from SCD to VCD.

**Last sentence in page 12: it would be beneficial to include a discussion with few sentences on the main findings even though they are published in Yu et al.**

As suggested a short summary of findings presented in Yu et al. 2021 has been included.

**Page 15 line 29: TROPOMI was not launch until October 2017.**

The year is actually not used to calculate the sun-observation geomtry, it is the same every year. Omitted "2017" in the text.

**Sect. 4.2.3: I would suggest to substitute the global maps of surface albedo for a zoom over the study for which the cloud simulations are done.**

Done.

**Sect. 4.2.4: What about higher resolution than 7 km x 7 km? This is good for TROPOMI, but future sensors will definitely provide measurements at higher resolution. In Sect. 3 the simulations are done for higher resolution, and the ICON clouds resolution is 1.2 km x 1.2 km. The increased spatial resolution (as pointed in the manuscript) will enhance the impact of 3D clouds effects, so it would benefit the discussion to perform these simulations at higher resolution. If this is not feasible, then at least this should be discussed.**

Higher spatial resolution would indeed be interesting. However, the aim of the 3DCATS study was to quantify the retrieval error for TROPOMI, therefore we generated the synthetic data for TROPOMI and not for future instruments. The impact of spatial resolution has been investigated using the box cloud synthetic data. This is discussed in Yu et al (part II). A reference to this has been added to the end of Section 3 of this paper.

**What is the effect on the results of reducing the number of photons with respect to the 1D/3D case? How will this affect the airmass factor calculation?**

The standard deviation of the Monte Carlo results increases, clarified this in the text:

"Note, that the number of photons was 100 times less than for the box-cloud and clear-sky cases presented in Section 3. Therefore, since the standard deviation of a Monte Carlo simulation is inversely proportional to the square root of the number of photons, the standard deviation of all simulations results (reflectance spectra and layer-AMFs) is increased by a factor of 10."

**The study uses a very realistic cloud field from LES simulations, but then assumes a constant NO2 field over the whole domain, which is very unrealistic. The consequences of this assumption on NO2 needs to be discussed. How would the NO2 retrieval error on Fig. 12 look like if a realistic NO2 field would be used?**

The retrieval error in Fig. 12 would look completely different if we use an inhomogeneous NO2 field, errors would be small in clean regions and larger in polluted regions. It would be very difficult to relate the results to the cloud properties. For this reason we decided to include a homogeneous NO2 profile in the background atmosphere. Anyway, we also provide the layer-AMFs which can be used to investigate different NO2 profiles.

We included the following statement as explanation: " We have chosen a constant $NO_2$ profile because we aim to investigate the impact of realistic clouds on the retrieval results. When we include an inhomogeneous $NO_2$ profile it is not easily possible to quantify this impact, e.g. to figure out, which type of clouds have the largest impact on the retrieval error. This is only possible when we have the same atmospheric background conditions over the full domain."

**Figures and figure captions should be revised. Different sub-figures are specified differently in different figures, so please revise. See https://www.atmospheric-measurement-techniques.net/submission.html for figure guidelines. Using letters**

**a,b,c etc. makes referencing on the text easier. Please mind the reader when creating the figures. For example, Fig. 10: '(similar to TROPOMI on Sentinel-5P and Sentinel-5)' this is not relevant in a figure caption. Lower panels 'x = 256 km' is not relevant information and makes the figure busier. Another example, Fig. 3 Top: legend 'clear' is better understood than '-1.5' and '-10.5 km', maybe add 'shadow' and 'clear region'.**

We revised the figures and included letters to refer to the panels in the figures. Further we revised several legends and figure titles as suggested.

**Name the O2 A band consistently throughout the manuscript (three names O2-A band, O2A-band, O2A band have been used)**

Named "$O_2$A-band" consistently.

**Page 2, line 28: for->from TROPOMI/S5P obs.**

Done.

**Page 2, line 29: synthetic -> synthetic**

Done.

**Page 2, line 30: are-> were not included**

Done.

**Page3, line 3: incorrect grammar; the bias due to 3D clouds on what? And no need to start new paragraph if you talk about the same paper.** Rephrased.

**Page 18, limes 5-10: please write sentences in present tense. E.g., "pathlength is decreased" -> "decreases"**

Done.

**Short paragraphs (1-2 sentences) just expressing technical details should be avoided.**

Merged the short paragraphs.

**Please include (at least) a reference when mentioning FRESCO cloud algorithm.**

Done.

**Page 22, line 6: what about the shadow effects?**

Rephrased: "... including effects of clouds (in-scattering and shadowing) ..."

**Page 22, line 23: was this not at 1.2 km x 1.2 km?**

Yes, thank you. Corrected this.

**References**

Buras, R. and Mayer, B.: Efficient unbiased variance reduction techniques for Monte Carlo simulations of radiative transfer in cloudy atmospheres: The solution, J. Quant. Spectrosc. Radiat. Transfer, 112, 434–447, 2011.

Deutschmann, T., Beirle, S., Frieß, U., Grzegorski, M., Kern, C., Kritten, L., Platt, U., Prados-Roman, C., Pukite, J., Wagner, T., Werner, B., and Pfeilsticker, K.: The Monte Carlo atmospheric radiative transfer model McArtim: Introduction and validation of Jacobians and 3D features, J. Quant. Spectrosc. Radiat. Transfer, 112, 1119 – 1137, https://doi.org/DOI:10.1016/j.jqsrt.2010.12.009, 2011.

Platt, U.: Air Monitoring by Differential Optical Absorption Spectroscopy, pp. 1–28, American Cancer Society, https://doi.org/https://doi.org/10.1002/9780470027318.a0706.pub2, URL https://onlinelibrary.wiley.com/doi/abs/10.1002/9780470027318.a0706.pub2, 2017.

Schwaerzel, M., Emde, C., Brunner, D., Morales, R., Wagner, T., Berne, A., Buchmann, B., and Kuhlmann, G.: Three-dimensional radiative transfer effects on airborne and ground-based trace gas remote sensing, Atmos. Meas. Tech., 13, 4277–4293, https://doi.org/10.5194/amt-13-4277-2020, URL https://amt.copernicus.org/articles/13/4277/2020/, 2020.

Yu, H., Emde, C., Kylling, A., van Roozendael, M., Stebel, K., Veihelmann, B., and Mayer, B.: Impact of 3D Cloud Structures on the Atmospheric Trace Gas Products from UV-VIS Sounders – Part II: impact on NO 2 retrieval and mitigation strategies, Atmos. Meas. Tech. Discuss., submitted, 2021.

---

## Author Comment (AC3)

**Answers to referee 1 comments "Impact of 3D Cloud Structures on the Atmospheric Trace Gas Products from UV-VIS Sounders – Part I: Synthetic dataset for validation of trace gas retrieval algorithms"**

We thank the reviewer for the constructive comments that helped us to improve the manuscript. In the following we respond point-by-point to all comments.

**This paper is one of a set of three interconnected papers that discusses a) a publicly available synthetic dataset of 3D radiances, b) the sensitivity of vertical column density NO2 retrieval errors near box-clouds and observations, and c) 3D cloud biases and metrics. The reviewed paper is part a) of the full set of papers. Since the paper is the first of a three-set collection, main findings are reserved for the other two papers. This produces the awkward situation that the main physical results, which one can derive from an analysis of the synthetic dataset, are not discussed in the reviewed paper. The reviewed paper is overly restrained. The reviewed paper is rather short of main findings, mainly stating that a synthetic data set is available, and therefore limited in informative results.**

Yes, the main scientific findings are included in part II and part III. However, we think, that the description of how the dataset is created, which methods are used etc. is worth a publication. To our knowledge this is the first synthetic dataset including high resolution spectra (as used for trace gas remote sensing) and three-dimensional realistic clouds, generated using a three-dimensional radiative transfer model.

**There are places in the text in which a terse one sentence paragraph is stated. Additional sentences can and should be added to the text in these portions of the text.**

We have included several additional explanations, e.g. we explained in more detail, how the layer-AMFs can be used to study effects of the $NO_2$ profile on the retrieval error:
"Using MYSTIC, we may study how the layer-AMFs are modified by scattering from clouds in the neighborhood. Comparing the layer-AMFs of a clear sky atmosphere with the layer-AMFs influenced by clouds, we may estimate the retrieval error of, e.g., $NO_2$ vertical column densities (VCDs). Working with simulated layer-AMFs allows us also to study the impact of the vertical $NO_2$ concentration profile on the retrieval error. Since the influence of trace gases on the photon pathlength distribution and thus on layer-AMF is negligible, we may use the layer-AMFs of one radiative transfer simulation to estimate the error for various assumed $NO_2$ concentration profiles."

**A suggested addition to the revised paper would be to include a figure or two that demonstrates the in-scattering and shadow curves (similar to Figure 2) for clouds of e.g. three heights in the LES cloud field, including panels in which (on the y axis) the reflectance is graphed as a function of distance from cloud edge, and panels in which retrieved NO2 is graphed (on the y axis) as a function of distance from cloud edge. These figures would help the research community better appreciate the quantitative**

**importance of 3D cloud effects upon NO2 retrievals. Over what km range are 3D effects present as a function of cloud height, and what are the column for these situations? I would have liked to have seen in the Conclusions section a discussion of the major physical findings of the paper. The Figures provide instructive insights, yet these insights are barely touched upon in the Conclusions.**

A detailed analysis using the synthetic dataset based on LES clouds is presented in the third paper (Kylling et al., 2021), which includes several figures showing the impact of 3D cloud scattering on the $NO_2$ VCD retrieval error. Since the paper is available also in AMT, we do not like to duplicate these figures.

We have included a summary of the scientific findings from this analysis to the conclusions:

"The dataset based on LES clouds has been used to quantify the $NO_2$ VCD retrieval error. An operational retrieval algorithm was applied on the synthetic observations and the retrieval results were compared against the true $NO_2$ VCD which is the known model input. The exemplary results show underestimations of the retrieved $NO_2$ VCD in cloud shadow regions of more than 20% and overestimations of about the same order of magnitude in in-scattering regions for the specific sun-observer geometry. In the third paper of the series (Kylling et al., 2021), an analysis of the complete dataset is presented. Cloud shadow fraction, cloud top height, cloud optical thickness, $NO_2$ profile, solar zenith and viewing angle have been identified as the most important parameters determining the impact of cloud scattering on the $NO_2$ VCD retrieval. For low-earth and geostationary orbit geometries, 89 and 93%, respectively, of the retrieved $NO_2$ VCD were within 10% of the actual VCD for solar zenith angles less than 60°. For a solar zenith angle of 60° the numbers decrease to 53 and 61%. It was also found that for solar zenith angles less than 10°, the $NO_2$ VCD retrieval error is generally smaller than 10%. For larger solar zenith angles the retrieval error increases to values of the order of tens of percent."

**Page 3, line 12. Clarify what is meant by "bias" (the bias of what?)**
**Page 3, line 12. Replace by "In the third paper by Kylling..". The one sentence paragraph is a bit jarring since it is overly short in informative content.**

Rephrased and merged 2 paragraphs summarizing the third paper of the series Kylling et al. (2021).

"In the third paper by Kylling et al. (2021) the $NO_2$ VCD retrieval error due to 3D cloud scattering has been quantified using both, synthetic and observational data."

**Page 4, line 3. Clarify what is meant by "unbiased radiances"**

Removed "unbiased" because it is not clear from the context.

There are variance reduction methods, e.g., the so-call phase function truncation method, which cause a bias in the computed radiances. The VROOM methods are physically correct, do not use approximations and therefore the results are not biased.

**Page 4, line 6. Replace "agreed perfectly" with a quantitative value.**

We provide six references to different model intercomparison studies. Depending on the specific study and on the investigated quantities, the level of agreement is not always the same, so we can not provide one quantitative value here, the reader needs to look into the given references. Removed the word "perfectly", because this might be a misleading term.

**Page 4, lines 14-17. I am not convinced of the ability of the authors to "calculate the full spectrum based on photon path distributions sampled at a single wavelength". Atmospheric optical properties (Rayleigh scattering, aerosol optical depths, asymmetry and single scattering albedo) have a wavelength dependence in a real atmosphere. Please support your statements in the context of a real atmosphere with additional sentences. The sentence "The statistical error of such a simulation is a bias for the complete spectrum" is not comprehensible. Add additional sentences which discuss the ALIS method. Replace with "This method allows one to calculate .."**

Included some additional sentences to describe the ALIS method. Please refer to Emde et al. (2011) to understand the details, this paper includes also a validation of the method.

"The Absorption Lines Importance Sampling (ALIS) method (Emde et al., 2011) solves this problem. This method allows one to calculate the full spectrum based on photon path distributions sampled at a single wavelength. In order to take into account the spectral dependence of the absorption coefficient a spectral absorption weight is calculated for each photon path. Further, at each scattering event the local estimate method (Marshak and Davis, 2005) is combined with an importance sampling method to take into account the spectral dependence of the scattering coefficient. Since each wavelength grid point is computed using the same photon path distribution, the statistical error of such a simulation is is almost independent of wavelength, i.e. it corresponds to a small offset of the complete spectrum. For DOAS type retrievals this error is completely removed by the polynomial fit to compute the differential optical thickness. This statistical error decreases with the number of photons used in the simulation and converges towards the correct spectrum. The method it is very well suited to efficiently simulate radiance spectra in high-spectral resolution."

**Page 5, line 1. What is the boundary layer height, and how is NO2 vertically distributed in the troposphere?**

Included a more detailed description of the model atmosphere (including vertical layering) and also a figure showing the NO2 profile.

**Page 5, line 10. Change to "in the x-direction".**

Done.

**Page 5, line 14. The sentence implies just a single box-cloud geometry, while Table 1 lists several box-cloud heights. Rephrase to "For the liquid water cloud the primary cloud geometry has the base height set to 2 km ...".**

This sentence refers to the base case, as stated in the sentence before.

Clarified later on that we start with the base case and vary the different parameters:

"Starting from the base case, we varied the following parameters: cloud optical thickness, cloud bottom height, cloud geometrical thickness, solar zenith angle, and surface albedo."

**Page 6, line 4. Why were aerosols not included in the calculations?**

We included the following explanation:

"Aerosols were not included, although aerosol scattering also has a significant impact on the $NO_2$ retrieval. However, in this study, we aim to quantify the impact on cloud scattering on the retrieval. When both, aerosols and clouds are included, it becomes difficult to disentangle the impacts of cloud and aerosol scattering. Therefore, we decided to include only clouds."

**Page 6, line 11. Clarify what is meant by "variance reduction methods". Add sentences that describe the VROOM methods.**

Included a short explanation:

"The variance reduction methods VROOM (Buras and Mayer, 2011), which reduce the statistical noise in Monte Carlo radiative transfer simulations including cloud scattering,"

Detailed explanations are out of scope of this paper, the reader is refered to Buras and Mayer (2011), which describes all methods thouroughly.

**Page 9, line 12. Change to "Note that D(l) is a smooth function"**
Done.

**Page 11, line 8. Change to "layer-AMF as a function of"**
Done.

**Page 13, line 2. Change to "realistic 3D clouds"**
Done.

**Page 15, line 17. Replace "sufficiently accurate" with a quantitative accuracy.**
Included "reflectances agree to 3 digits after the decimal point ".

**Page 15, line 22. Change to "was analyzed, and it was found that SZA.."**
Done.

**Page 16, line 1. Change to "It was found that SZA varies.."**
Done

**Page 17, line 1. Change to "Note that each simulated pixel includes 36 cloud pixels,.."**
Done.

**Page 17, line 5. Change to "Note that the number of .."**

Done.

**Page 18, Figure 11. What are the units of the NO2 retrieval error?**

The figure shows the relative error (no unit).

**Page 19, line 16. Change to "Note that the complete LES.."**

Done.

**Page 23, line 7. Explain how the synthetic dataset can be used to "validate the various different trace gas retrieval approaches for Sentinel-S5P." This is an example of a terse one sentence paragraph that would benefit from additional sentences.**

Included additional sentences: "The algorithms can be applied on the synthetic data. Comparing the retrieved $NO_2$ VCDs to the true value used as input to the radiative transfer simulations yields the retrieval accuracy of each algorithm."

**References**

Buras, R. and Mayer, B.: Efficient unbiased variance reduction techniques for Monte Carlo simulations of radiative transfer in cloudy atmospheres: The solution, J. Quant. Spectrosc. Radiat. Transfer, 112, 434–447, 2011.

Emde, C., Buras, R., and Mayer, B.: ALIS: An efficient method to compute high spectral resolution polarized solar radiances using the Monte Carlo approach, J. Quant. Spectrosc. Radiat. Transfer, 112, 1622–1631, 2011.

Kylling, A., Emde, C., Yu, H., van Roozendael, M., Stebel, K., Veihelmann, B., and Mayer, B.: Impact of 3D Cloud Structures on the Atmospheric Trace Gas Products from UV-VIS Sounders – Part III: bias estimate using synthetic and observational data, Atmos. Meas. Tech. Discuss., submitted, 2021.

Marshak, A. and Davis, A.: 3D Radiative Transfer in Cloudy Atmospheres, Springer, iSBN-13 978-3-540-23958-1, 2005.

---

## Referee Report (RR1)

Review of "Impact of 3D Cloud Structures on the Atmospheric Trace Gas Products from UV-VIS Sounders – Part I: Synthetic dataset for validation of trace gas retrieval algorithms" by Emde et al.(AMT-2021-336)

This paper is one of a set of three interconnected papers that discusses a) a publicly available synthetic dataset of 3D radiances, b) the sensitivity of vertical column density $NO_2$ retrieval errors near box-clouds and observations, and c) 3D cloud biases and metrics. The reviewed paper is part a) of the full set of papers. The remote sensing community can obtain and use the synthetic dataset of 3D radiances by accessing a Zenodo web site.

The authors have responded adequately to the comments and suggested revisions to the original paper.

The paper should be published following minor suggested text changes.

General comments

The paper contains very informative Figures. I especially like Figures 3,4,5 which looks at 1D and 3D calculations for idealized clouds.

The realistic LES cloud scene is well chosen to illustrate the complexity of actual atmospheric scenes.

The paper forms a carefully written series of three papers which will be well received by the remote sensing community.

Minor suggested changes to the text

Page 2, line 9 change to "and it was found that cloud"

Page 2, line 12 change to "or photon path length correction"

Page 2, line 13. Provide several references where the three effects ae included in operational cloud correction methods.

Page 2, line 20 change to "studies have shown that 3D cloud"

Page 3, line 4 change to "box-clouds. Yu et al (2021) systematically analyzes the VCD retrieval error in terms of the following"

Page 3, line 9 change to "using both synthetic and"

Page 3, line 11 change to "the first part of the synthetic data"

Page 3, line 25 change to "as Lambertian or by a Bidirectional"

Page 3, line 31 change to "and always agreed well to other participating radiative transfer codes"

Page 4, line 17 change to "DOAS technique (Platt, 2017):"

Page 4, line 28 change to "MYSTIC calculates layer-AMFS'

Page 5, line 1 change to "partial column density (of $NO_2$) for layer"

Page 5, line 20 change to "atmosphere of Anderson"

Page 5, line 22 change to "cm$^2$, with most of"

Page 13, line 4 change to "with the 1D cloud layer included"

Page 13, line 12 change to "or by gas molecules"

Page 16, line 5. Define the H-metric and provide a reference.

Page 17, line 6 change to "to figure out which type of "

Page 17, line 18 change to "and it was found that SZA"

Page 23, line 13 change to "region of the O$_2$"

Page 24, line 17 change to "to ensure that the LES"

1. Does the paper address relevant scientific questions within the scope of AMT? yes

2. Does the paper present novel concepts, ideas, tools, or data? The box-cloud and LES 3D calculations are fairly unique

3. Are substantial conclusions reached? Substantial conclusions are mainly contained in the 2$^{nd}$ and 3$^{rd}$ papers of the three paper set. The discussion in the reviewed paper presents very informative graphs and text which will be helpful to educate the remote sensing community.

4. Are the scientific methods and assumptions valid and clearly outlined? Some additional sentences could have been added to the text to clarify some of the methods. The text does have adequate referencing.

5. Are the results sufficient to support the interpretations and conclusions? yes

6. Is the description of experiments and calculations sufficiently complete and precise to allow their reproduction by fellow scientists (traceability of results)? yes

7. Do the authors give proper credit to related work and clearly indicate their own new/original contribution? yes

8. Does the title clearly reflect the contents of the paper? yes

9. Does the abstract provide a concise and complete summary? yes

10. Is the overall presentation well structured and clear? The paper is well organized.

11. Is the language fluent and precise? Yes (with the exception of a few sentences)

12. Are mathematical formulae, symbols, abbreviations, and units correctly defined and used? yes

13. Should any parts of the paper (text, formulae, figures, tables) be clarified, reduced, combined, or eliminated? The paper is long enough, so additional text could become burdensome.

14. Are the number and quality of references appropriate? yes

15. Is the amount and quality of supplementary material appropriate? yes

---

## Author Response (AR2)

**Answers to referee comments "Impact of 3D Cloud Structures on the Atmospheric Trace Gas Products from UV-VIS Sounders – Part I: Synthetic dataset for validation of trace gas retrieval algorithms"**

We thank the reviewer for providing further comments and suggestions.

We included all text changes as suggested:

**Page 2, line 9 change to "and it was found that cloud"**
Done.

**Page 2, line 12 change to "or photon path length correction"**
Done.

**Page 2, line 13. Provide several references where the three effects ae included in operational cloud correction methods.**
We refer here to the references mentioned in the sentence before. This has now been clarified.

**Page 2, line 20 change to "studies have shown that 3D cloud"**
Done.

**Page 3, line 4 change to "box-clouds. Yu et al (2021) systematically analyzes the VCD retrieval error in terms of the following"**
Done.

**Page 3, line 9 change to "using both synthetic and"**
Done.

**Page 3, line 11 change to "the first part of the synthetic data"**
Done.

**Page 3, line 25 change to "as Lambertian or by a Bidirectional"**
Done.

**Page 3, line 31 change to "and always agreed well to other participating radiative transfer codes"**
Done.

**Page 4, line 17 change to "DOAS technique (Platt, 2017):"**
Done.

**Page 4, line 28 change to "MYSTIC calculates layer-AMFS"**
Done.

**Page 5, line 1 change to "partial column density (of NO2) for layer"**
Done.

**Page 5, line 20 change to "atmosphere of Anderson"**
Done.

**Page 5, line 22 change to "cm2, with most of"**
Done.

**Page 13, line 4 change to "with the 1D cloud layer included"**
Done.

**Page 13, line 12 change to "or by gas molecules"**

Done.

**Page 16, line 5. Define the H-metric and provide a reference.**
Thank you for this comment. We have now included the definition of the H-metric:

"For the calculation of the H-metric of a $7{\times}7\,\text{km}^2$ area corresponding to the size of a TROPOMI pixel we take into account 36 simulated band reflectences contained in this area. The H-metric is the standard deviation of these reflectances divided by their mean value and it provides an estimate of the variation of reflectance within a TROPOMI pixel (Kylling et al., 2021)."

**Page 17, line 6 change to "to figure out which type of "**
Done.

**Page 17, line 18 change to "and it was found that SZA"**
Done.

**Page 23, line 13 change to "region of the O2"**
Done.

**Page 24, line 17 change to "to ensure that the LES"**
Done.

**References**

Kylling, A., Emde, C., Yu, H., van Roozendael, M., Stebel, K., Veihelmann, B., and Mayer, B.: Impact of 3D Cloud Structures on the Atmospheric Trace Gas Products from UV-VIS Sounders – Part III: bias estimate using synthetic and observational data, Atmos. Meas. Tech. Discuss., submitted, 2021.